# Methylglyoxal, a glycolysis side-product, induces Hsp90 glycation and YAP-mediated tumor growth and metastasis

Marie-Julie Nokin[1], Florence Durieux[1], Paul Peixoto[1], Barbara Chiavarina[1], Olivier Peulen[1], Arnaud Blomme[1], Andrei Turtoi[1], Brunella Costanza[1], Nicolas Smargiasso[2], Dominique Baiwir[3], Jean L Scheijen[4], Casper G Schalkwijk[4,5], Justine Leenders[6], Pascal De Tullio[6], Elettra Bianchi[7], Marc Thiry[8], Koji Uchida[9], David A Spiegel[10], James R Cochrane[11], Craig A Hutton[11], Edwin De Pauw[2], Philippe Delvenne[7], Dominique Belpomme[12], Vincent Castronovo[1], Akeila Bellahcène[1]*

[1]Metastasis Research Laboratory, GIGA-CANCER, University of Liège, Liège, Belgium; [2]Mass Spectrometry Laboratory, GIGA-Systems Biology and Chemical Biology, University of Liège, Liège, Belgium; [3]GIGA Proteomic Facility, University of Liège, Liège, Belgium; [4]Laboratory for Metabolism and Vascular Medicine, Department of Internal Medicine, Maastricht University, Maastricht, Netherlands; [5]Cardiovascular Research Institute Maastricht, Maastricht, The Netherlands; [6]Laboratory of Medicinal Chemistry - CIRM, University of Liège, Liège, Belgium; [7]Department of Pathology, CHU, University of Liège, Liège, Belgium; [8]Laboratory of Cellular and Tissular Biology, GIGA-Neurosciences, University of Liège, Liège, Belgium; [9]Laboratory of Food and Biodynamics, Graduate School of Bioagricultural Sciences, University of Nagoya, Nagoya, Japan; [10]Department of Chemistry, Yale University, New Haven, United States; [11]School of Chemistry and Bio21 Molecular Science and Biotechnology Institute, University of Melbourne, Melbourne, Australia; [12]Association for Research and Treatments Against Cancer, Paris, France

*For correspondence:
a.bellahcene@ulg.ac.be

Competing interests: The authors declare that no competing interests exist.

**Abstract** Metabolic reprogramming toward aerobic glycolysis unavoidably induces methylglyoxal (MG) formation in cancer cells. MG mediates the glycation of proteins to form advanced glycation end products (AGEs). We have recently demonstrated that MG-induced AGEs are a common feature of breast cancer. Little is known regarding the impact of MG-mediated carbonyl stress on tumor progression. Breast tumors with MG stress presented with high nuclear YAP, a key transcriptional co-activator regulating tumor growth and invasion. Elevated MG levels resulted in sustained YAP nuclear localization/activity that could be reverted using Carnosine, a scavenger for MG. MG treatment affected Hsp90 chaperone activity and decreased its binding to LATS1, a key kinase of the Hippo pathway. Cancer cells with high MG stress showed enhanced growth and metastatic potential in vivo. These findings reinforce the cumulative evidence pointing to hyperglycemia as a risk factor for cancer incidence and bring renewed interest in MG scavengers for cancer treatment.

## Introduction

Unlike normal cells, cancer cells mainly rely on glycolysis to generate energy needed for cellular processes even in normoxia conditions. This process referred to aerobic glycolysis or the 'Warburg

effect' is considered as a hallmark of cancer cells (*Ward and Thompson, 2012*). Although aerobic glycolysis is less efficient than respiration to generate ATP, we know now that it effectively supports the anabolic requirements associated with cancer cell growth and proliferation. One underestimated consequence of increased glucose uptake and glycolytic flux is the accumulation of potent toxic metabolites such as reactive carbonyl species. Among those, methylglyoxal (MG) is a highly reactive α-oxoaldehyde that is primarily formed in cells by the spontaneous degradation of triose phosphate intermediates of glycolysis, dihydroxyacetone phosphate and glyceraldehyde 3-phosphate (*Richard, 1993*). Alpha-oxoaldehydes are up to 20,000-fold more reactive than glucose in glycation processes (*Turk, 2010*), and it is expected that 1% to 5% of proteins in cells are modified by MG (*Rabbani and Thornalley, 2014*). MG leads to chemical modification of proteins, lipids and nucleotides that result in cellular dysfunction and mutagenicity. MG interaction with amino groups of proteins notably leads to the formation of advanced glycation end products (AGEs) called hydroimidazolones (MG-H) and argpyrimidines (*Thornalley, 1996*). All mammalian cells possess a detoxifying system constituted of glyoxalases 1 and 2 (Glo1 and Glo2, respectively), which catalyze the conversion of MG to D-lactate (*Thornalley, 2005*). The disturbance in the balance between endogenous reactive carbonyl species generation and the ability to counteract their harmful effects is defined as the carbonyl stress.

At the molecular level, carbonyl stress is a common feature of the metabolic dysfunction associated with diabetes and cancer. MG-related AGEs have been found to be increased two- to fivefold and have been mainly identified in the context of diabetes. For example, MG post-translational modification of vascular basement membrane type IV collagen (*Dobler et al., 2006*) and of voltage-gated sodium channel Nav1.8 (*Bierhaus et al., 2012*) have been associated with long-term diabetic complications.

Although the link between oxidative stress, cancer development, progression and response to therapy is clearly established, carbonyl stress and cancer connection remains largely unexplored and has never been envisaged as potentially interconnected. To the best of our knowledge, only one study has reported MG-derived AGEs detection in malignant tumors (*van Heijst et al., 2005*). Using immunohistochemistry, we recently reported the accumulation of argpyrimidine MG adducts in breast cancer tumors (*Chiavarina et al., 2014*). Remarkably, MG-mediated glycation of specific target proteins happens to be beneficial to cancer progression. For example, the formation of argpyrimidine on heat-shock protein 27 (Hsp27) prevented cancer cell apoptosis in lung (*van Heijst et al., 2006*) and gastrointestinal (*Oya-Ito et al., 2011*) cancers. Moreover, inhibition of MG modification on Hsp27 caused sensitization of cancer cells to antitumoral drugs (*Sakamoto et al., 2002*).

MG has been shown to down regulate Hsp90 and Hsc70 expression levels in human retinal pigment epithelial cells (*Bento et al., 2010*). Hsp90 is a molecular chaperone that gained great interest over the last 20 years as a druggable target for cancer treatment. Hsp90 stabilizes and activates more than 400 proteins, referred to as Hsp90 'clients', many of which are oncoproteins including transcription factors and kinases that are essential for cellular signal transduction pathways and adaptive responses to stress (*Trepel et al., 2010*). One such client protein is the large tumor suppressor 1 (LATS1) (*Huntoon et al., 2010*), a key kinase that relays anti-proliferative signals in the Hippo pathway through Yes-associated protein (YAP) and transcriptional co-activator with PDZ-binding motif (TAZ) phosphorylation and inactivation (*Pan, 2010*). Consistent with its fundamental role in the control of organ growth and size in vertebrates, the dysfunction of the Hippo signalization triggers tumorigenesis in human (*Harvey et al., 2013*). As a co-activator of TEAD family of transcription factors (*Zhao et al., 2008*), YAP has been notably shown to enhance cancer progression through transcriptional activation of proliferation promoting genes such as c-myc and CTGF (*Moroishi et al., 2015*). Recent studies established a link between glucose deprivation stress, aerobic glycolysis and YAP activation in cancer (*DeRan et al., 2014*; *Enzo et al., 2015*; *Mulvihill et al., 2014*). Thus, reinforcing the increasing evidence indicating that metabolic pathways play causative roles in conferring an aggressive phenotype upon cancer cells. Because spontaneous MG accumulation results from the glycolytic flux, we hypothesized that MG stress might couple glycolysis to YAP activity. In this study, we show that MG induces YAP nuclear persistence and activity in breast cancer cells and we validate a molecular mechanism implicating MG-mediated Hsp90 inactivation and subsequent LATS1 kinase decrease. Our study establishes for the first time the functional significance of endogenous MG stress and reveals its unexpected connection with cancer cells propensity to grow and metastasize.

## Results

### Methylglyoxal adducts and nuclear YAP are positively correlated in human breast cancer

At the molecular level, a predictable consequence of the glycolytic switch in cancer cells is the induction of carbonyl stress (*Figure 1A*). We have previously reported MG-mediated carbonyl stress, assessed by argpyrimidine adducts detection, in a series of breast cancer lesions (*Chiavarina et al., 2014*). Recent reports highlighted the importance of glucose metabolism for the regulation of YAP activity in cancer cells (*DeRan et al., 2014*; *Enzo et al., 2015*; *Mulvihill et al., 2014*). To explore

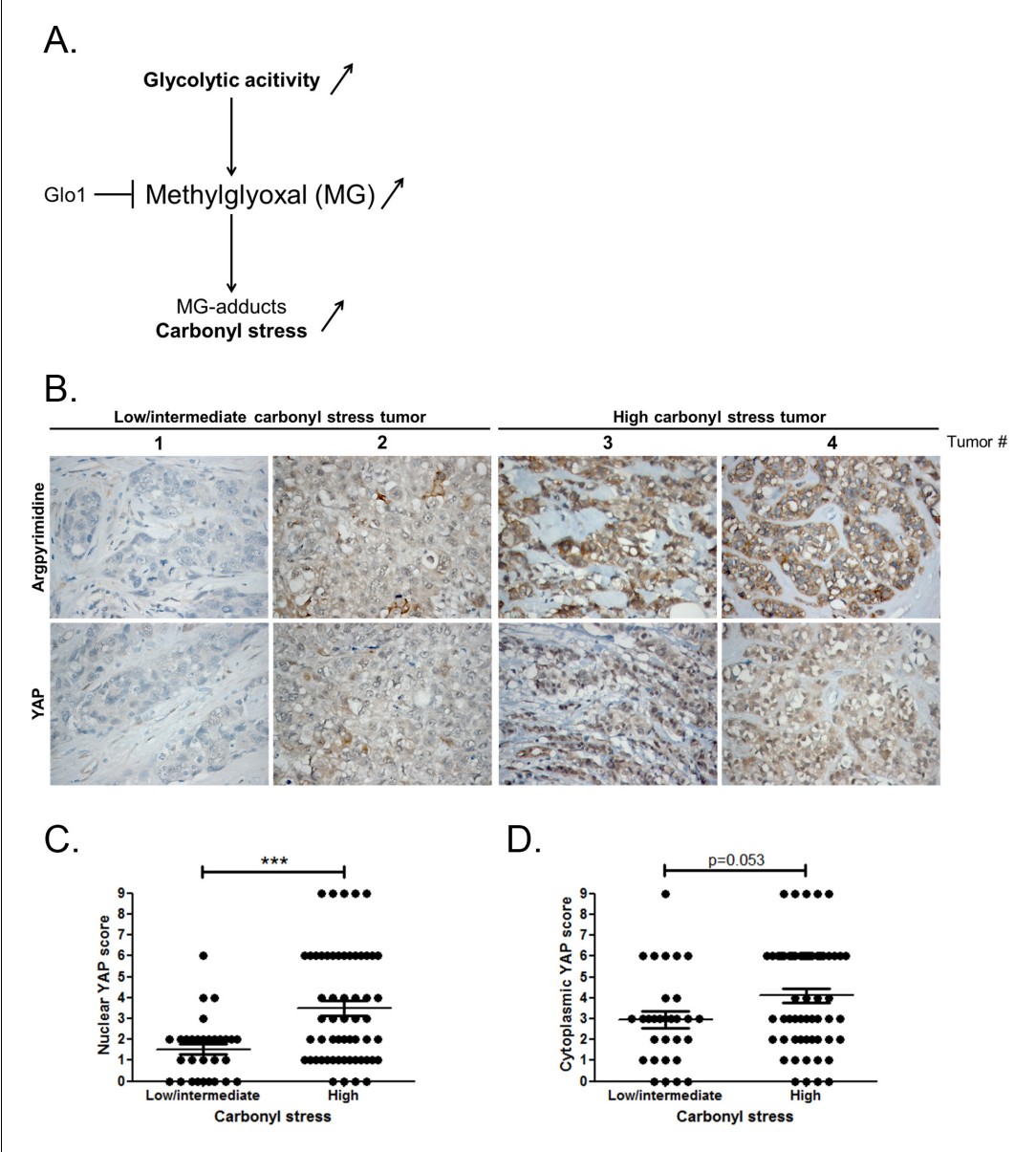

**Figure 1.** High carbonyl stress and nuclear YAP are positively associated in human breast cancer. (A) In cancer cells, a high glycolytic metabolism and/or a decrease of the MG-detoxifying activity of Glyoxalase 1 (Glo1) lead to high MG level thus establishing a carbonyl stress. (B) YAP IHC staining in representative low/intermediate and high carbonyl stress human breast tumors as assessed by their argpyrimidine level. (C) Quantification of nuclear and (D) cytoplasmic YAP IHC staining in a series of human breast cancer (n = 87). Each dot represents one case and bars represent mean ± SEM. Data were analyzed using Mann Whitney U and Wilcoxon Rank Sum tests and *** represents p<0.001.

possible links between YAP activity and carbonyl stress, we performed immunohistochemistry staining of YAP on a series of 87 breast tumors categorized as high and low-to-intermediate carbonyl stress tumors based on their endogenous argpyrimidine level. Remarkably, breast cancer lesions with high carbonyl stress also showed high YAP expression (*Figure 1B*). YAP was scored for nuclear and cytoplasmic staining. Statistical analysis revealed a significant difference between nuclear YAP staining in low/intermediate and high carbonyl stress tumors (*Figure 1C*). We demonstrated a positive correlation ($R_{spearman}$ = 0.3975, p=0.0001) between carbonyl stress intensity and nuclear YAP detection. Cytoplasmic YAP staining showed no significant difference between high and low/intermediate carbonyl stress breast tumors (*Figure 1D*).

## Methylglyoxal induces YAP persistence in confluent breast cancer cells

Deficient contact inhibition is a hallmark of invasive cancer cells, yet unexpectedly the density at which cancer cells are cultured impacts on the Hippo pathway in commonly studied cancer cell lines. In order to explore further the potential connection existing between MG-induced carbonyl stress and YAP, we first examined cell-density-dependent YAP subcellular localization in MDA-MB-231, MDA-MB-468 and MCF7 breast cancer cell lines. YAP was mainly localized in the nucleus of low-density cultured cancer cells as detected by immunofluorescence. When breast cancer cells reached confluence, YAP was not detectable in the nucleus and became generally less visible suggesting that it underwent degradation (*Figure 2A* and *Figure 2—figure supplement 1A and D*). Upon MG treatment, MDA-MB-231 cells showed a concentration dependent persistence of YAP in both the cytoplasm and the nucleus despite the cells reached confluence (*Figure 2A*). As a transcriptional coactivator, YAP's function is strictly constrained by its subcellular localization, thus we will essentially focus on YAP nuclear localization thereafter. Quantification supported that nuclear YAP immunodetection was dose dependently higher in MG treated cells when compared to untreated cells in high-density cultures (*Figure 2B*). Nuclear YAP accumulation was also found to be significant in MDA-MB-468 and MCF7 breast cancer cells upon 300 and 500 µM MG treatments (*Figure 2—figure supplement 1A,B,D, and E*). We obtained similar results in all three cell lines using a second antibody specifically directed against YAP (*Figure 2—figure supplement 2A,B and C*). We next showed that TAZ, the YAP paralog in mammalian cells, was modulated in the same way in breast cancer cells under MG treatment (*Figure 2—figure supplement 3*). Analysis of total YAP and TAZ expression using Western blot further demonstrated their increase in MG-treated cancer cells (*Figure 2C* and *Figure 2—figure supplement 1C and F*). A decreased or a stable cytoplasmic P-YAP (S127 and S381) level was observed and was consistent with nuclear YAP accumulation upon MG treatment (*Figure 2C* and *Figure 2—figure supplement 1C and F*). YAP mRNA levels were not significantly changed upon MG treatment in the three breast cancer cell lines (*Figure 2—figure supplement 1G*).

Data gathered so far indicates that MG favors YAP persistence in cancer cells. Next, we asked whether the blockade of MG-mediated carbonyl stress using carnosine, a known MG scavenger (*Hipkiss and Chana, 1998*), could abolish these effects. When MDA-MB-231 cells were concomitantly treated with MG and carnosine, YAP cellular accumulation in high-density cultures was significantly returned to untreated cells basal level (*Figure 2D and E*) indicating that YAP persistence in confluent cells directly or indirectly resulted from MG-mediated carbonyl stress. Carnosine alone did not affect significantly cellular YAP immunodetection. After we have validated exogenous MG effects, we used 2 strategies in order to assess high endogenous MG impact on YAP in breast cancer cells: (a) inhibition of *GLO1*, the main MG-detoxifying enzyme and (b) high-glucose culture condition.

## High endogenous methylglyoxal induces nuclear YAP accumulation in breast cancer cells

First, *GLO1* inhibition was achieved by the use of siRNAs on one hand and the use of S-p-bromobenzylglutathione cyclopentyl diester (BBGC), an effective Glo1 inhibitor on the other hand [*Tikellis et al., 2014*]. MBo, a specific fluorescent sensor for MG in live cells [*Wang et al., 2013*], demonstrated endogenous MG increase upon Glo1 expression inhibition and BBGC treatment in MDA-MB-231 cells (*Figure 3A*). Consistent with exogenous MG treatment experiments, both *GLO1*-depleted and BBGC-treated MDA-MB-231 cells (*Figure 3A and B*) displayed nuclear YAP

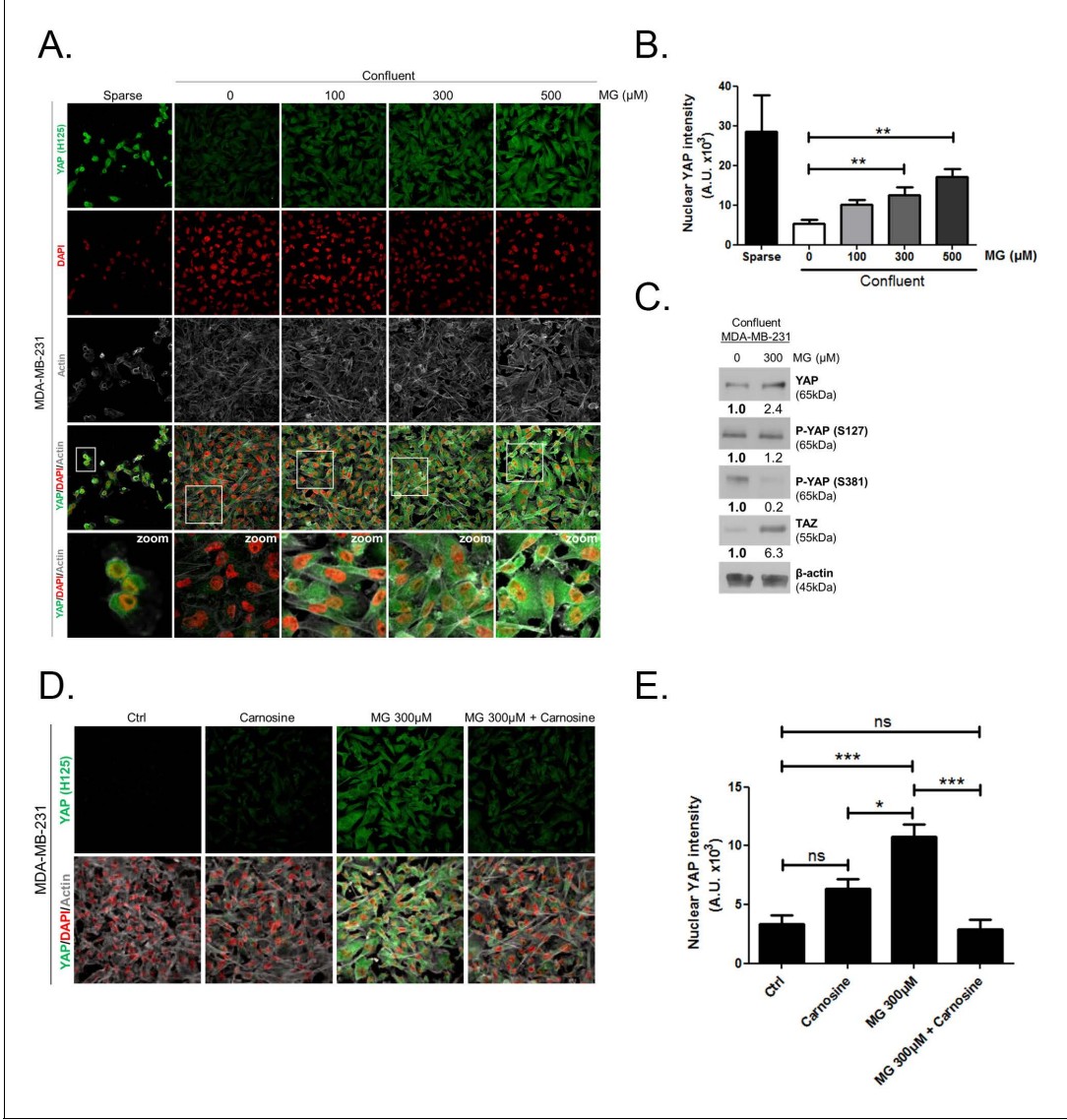

**Figure 2.** Methylglyoxal induces YAP persistence in confluent breast cancer cells. (A) Immunofluorescence (IF) staining shows that YAP (Santa Cruz antibody, H125) is mainly localized in the nucleus at low cellular density (Sparse) and is weakly detectable at high cellular density (Confluent) in MDA-MB-231 cells. In contrast, cells treated with increasing doses of MG until they reach confluence showed significant YAP cellular accumulation. Zoomed pictures are shown where indicated. Magnification 630x. Data are representative of three independent experiments. (B) Quantification of panel A experiment reports the intensity of YAP staining that colocalized with DAPI staining as described in 'Materials and methods' section. Nuclear YAP IF staining intensity shows a significant dose-dependent increase in presence of MG. Data were analyzed using one-way ANOVA followed by Dunnett post-test and shown as the mean values ± SEM of three independent experiments. (C) YAP, P-YAP (S127 and S381) and TAZ expression in MDA-MB-231 cells treated with MG (300 μM) until they reached confluence using western blot. Immunoblot data were quantified by densitometric analysis and normalized for β-actin. Numbers represent fold increase relative to the condition shown with bold number. (D) MDA-MB-231 cells cultured until they reached high density and treated concomitantly with MG (300 μM) and carnosine (10 mM), a MG scavenger, impeded cellular accumulation of YAP. Magnification 630x. Data are representative of three independent experiments. (E) Quantification of panel D experiment. Data were analyzed using one-way ANOVA followed by Bonferroni post-test and are shown as the mean values ± SEM of three independent experiments. *p<0.05, **p<0.01, ***p<0.001 and ns = not significant.

The following figure supplements are available for figure 2:

**Figure supplement 1.** Methylglyoxal induces YAP accumulation in confluent breast cancer cells.

**Figure supplement 2.** Methylglyoxal induces YAP accumulation in confluent breast cancer cells.

*Figure 2 continued on next page*

*Figure 2 continued*

**Figure supplement 3.** Methylglyoxal induces TAZ accumulation in confluent breast cancer cells.

persistence in high-density cultures. Similar results were obtained under both conditions in MDA-MB-468 cells (*Figure 3—figure supplement 1A and B*). Efficient *GLO1* silencing in breast cancer cells was assessed by Glo1 immunoblotting (*Figure 3—figure supplement 1C and D*). Altogether, these results showed that MG stress maintained detectable YAP nuclear levels in confluent breast cancer cells.

Second, we cultured MDA-MB-231 (highly glycolytic) and MCF7 (low glycolytic) cells in low- and high-glucose medium. Lactate measurement using $^1$H-NMR showed that MDA-MB-231 cells significantly increased their glycolytic activity when cultured in high glucose compared to low glucose (*Figure 3C*). In these cells, high-glucose culture induced elevated endogenous MG level that was assessed using both FACS detection of MBo fluorescent probe (*Figure 3D*) and LC-MS/MS quantification (*Figure 3E*). Similar results were observed in the other highly glycolytic breast cancer cell line, MDA-MB-468 (*Figure 3—figure supplement 1E–G*). As expected, low glycolytic MCF7 cells used for comparison did not react to high-glucose culture condition and kept stable lactate (*Figure 3C*). More importantly, MCF7 cells showed stable MG levels (*Figure 3D and E*) thus pointing for the first time to MG increase as a specific response of glycolytic cancer cells to glucose stimulus. After having validated the response of breast cancer cells to high glucose, we next asked whether YAP and TAZ nuclear persistence occurred under glucose-induced elevated endogenous MG levels. MDA-MB-231 and MDA-MB-468 cells cultured to confluence in high glucose demonstrated positive nuclear YAP and TAZ staining (*Figure 3F and G* and *Figure 3—figure supplement 1H and I*; and *Figure 3—figure supplement 2*) when compared with cells cultured in low glucose. Next, we reasoned that the inhibition of the glycolytic flux using the glycolysis inhibitor 2-deoxyglucose (2-DG) would reverse this effect. We first validated the decrease of lactate and MG production upon 2-DG treatment using $^1$H-NMR and FACS detection of MBo fluorescent probe, respectively (*Figure 3—figure supplement 3A and B*). As expected, YAP accumulation was not detectable in high-glucose MDA-MB-231 and MDA-MB-468 cells treated with 2-DG just like in low-glucose cultured cells (*Figure 3—figure supplement 3C and D*).

As expected from their stable glycolytic rate and unaffected MG level (*Figure 3C,D and E*), we did not observe any significant persistence of YAP and TAZ in MCF7 breast cancer cells (*Figure 3H and I* and *Figure 3—figure supplement 2*). It is noteworthy that MCF7 cells are able to induce YAP accumulation in response to an exogenous MG supply (*Figure 2—figure supplement 1*) suggesting that low glycolytic cells could be stimulated in a high MG environment created by neighboring cells for example and this, independently of their own glycolytic flux.

Finally, the observed effects of endogenous high MG levels on YAP were significantly reversed using 2 MG scavengers, carnosine and aminoguanidine in MDA-MB-231 cells (*Figure 3—figure supplement 4*). Altogether, these data demonstrate that the glycolytic switch in cancer cells is accompanied by high MG levels and YAP nuclear persistence thus establishing a new link between glucose utilization, MG stress and YAP regulation in cancer cells.

## MG induces YAP co-transcriptional activity in breast cancer cells

We next explored the functional relevance of MG-mediated nuclear accumulation of YAP in breast cancer cells. For this purpose, we used two shRNAs specifically directed against *GLO1* to stably induce high endogenous MG stress in MDA-MB-231 breast cancer cells. Efficient *GLO1* silencing (shRNAs #1 and #2) at the mRNA and protein levels and decreased Glo1 activity were validated in stably depleted clones (*Figure 4A,B and C*, respectively). As expected, *GLO1*-depleted MDA-MB-231 cells showed YAP and TAZ accumulation in high-cell-density cultures when compared to control cells (*Figure 4D and E* and *Figure 4—figure supplement 1A*). Consistently, this effect was significantly reversed using carnosine and aminoguanidine MG scavengers (*Figure 4—figure supplement 1B*). Stably depleted *GLO1* MDA-MB-231 cells were used to assess YAP target genes expression based on a previously established gene signature denoting YAP/TAZ activity (*Zhao et al., 2008*; *Cordenonsi et al., 2011*; *Dupont et al., 2011*; *Zhang et al., 2009*). Among the 14 targets tested

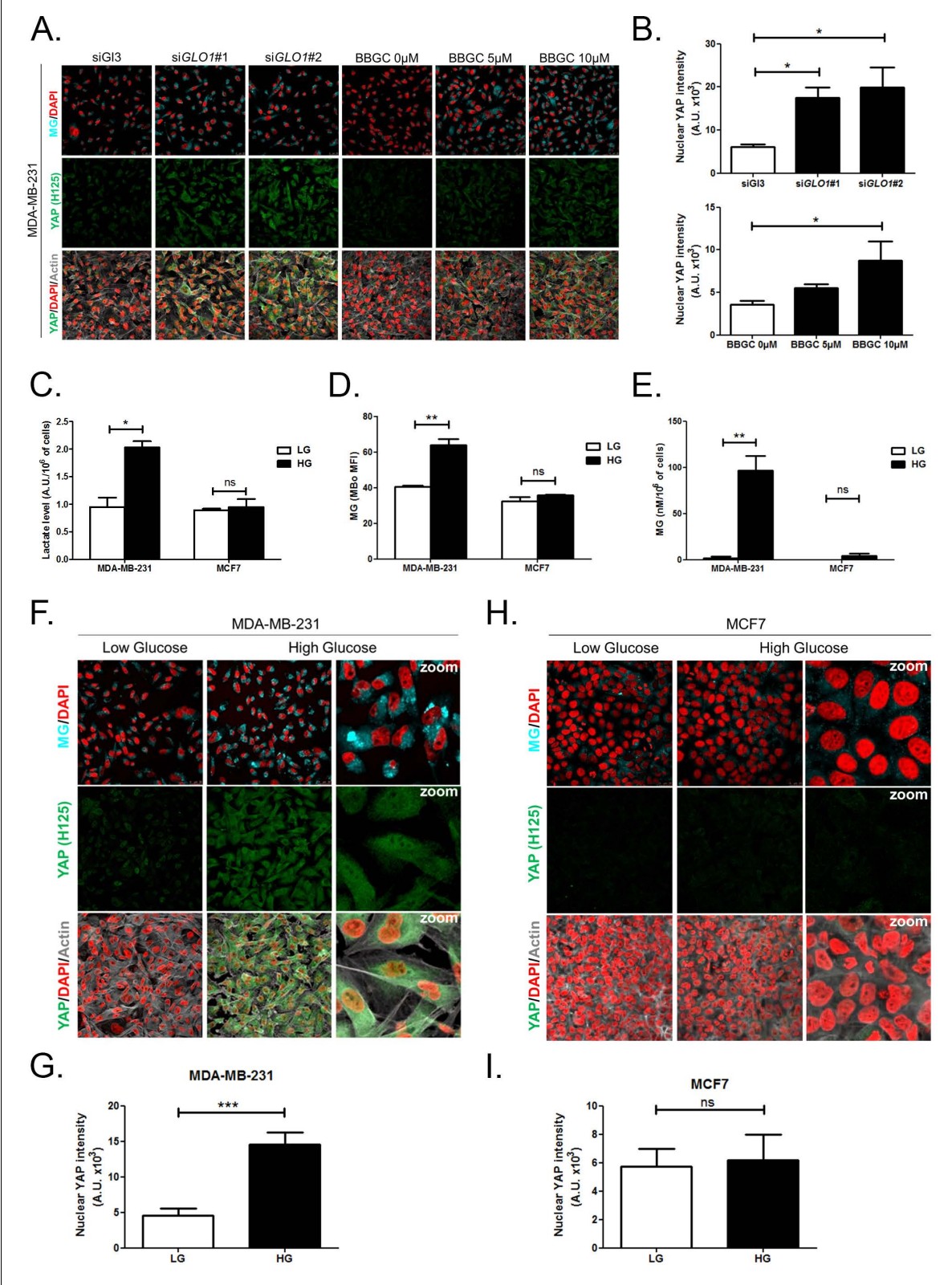

**Figure 3.** High endogenous MG induces YAP nuclear accumulation in breast cancer cells. (**A**) Detection of MG was performed using MBo specific fluorescent probe, as described in Materials and Methods section, and showed MG cellular increase in MDA-MB-231 cells that were *GLO1*-depleted using siRNAs (si*GLO1*#1 and #2) or treated with BBGC Glo1 activity inhibitor. Upon *GLO1* silencing/inhibition, MDA-MB-231 cells displayed more YAP (Santa Cruz antibody, H125) than control cells (siGl3 and BBGC 0 µM, respectively). Magnification 630x. Data are representative of three independent

*Figure 3 continued on next page*

*Figure 3 continued*

experiments. (**B**) Quantification of panel A experiment reports the intensity of YAP staining that colocalized with DAPI staining as described in Materials and Methods section for *GLO1* silencing and BBGC conditions. Data were analyzed using one-way ANOVA followed by Dunnett post-test and shown as the mean values ± SEM of three independent experiments. (**C**) Lactate level measured using $^1$H-NMR increased in highly glycolytic MDA-MB-231 cells cultured in high glucose (HG) compared to low glucose (LG) while MCF7 low glycolytic cells did not. (**D** and **E**) MG quantification using both FACS MBo mean fluorescence intensity (MFI) and LC-MS/MS analysis on conditioned medium in the indicated conditions as described under 'Materials and methods' section. MDA-MB-231 cells significantly increased their MG production in HG when compared to MCF7. (**F** and **H**) MG detection and YAP immunofluorescence staining (Santa Cruz antibody, H125) in the indicated breast cancer cell line cultured in low- and high-glucose medium. Magnification 630x. Zoomed pictures are shown for high-glucose condition. Data are representative of three independent experiments. (**G** and **I**) Quantification of F and H panels, respectively. Data shown in C, D, E, G, and I. were analyzed using unpaired Student's t test for each cell line independently and shown as the mean values ± SEM of three independent experiments. *p<0.05, **p<0.01, ***p<0.001 and ns = not significant.

The following figure supplements are available for figure 3:

**Figure supplement 1.** High endogenous MG induces YAP localization in breast cancer cells.

**Figure supplement 2.** High endogenous MG induces TAZ localization in breast cancer cells.

**Figure supplement 3.** Inhibition of glycolysis by treatment with 2-Deoxyglucose (2-DG) reverses YAP accumulation in MDA-MB-231 and MDA-MB-468 cells cultured in high-glucose medium.

**Figure supplement 4.** Carnosine and aminoguanidine MG scavengers reverse YAP accumulation in MDA-MB-231 cells cultured in high-glucose medium.

and known to be regulated positively by YAP, we found that 8 genes, including CTGF gene, showed a significant increase at the mRNA level in *GLO1* depleted cells when compared to control (*Figure 5A*). Importantly, knock-down of YAP using siRNA transfection reversed, at least in part, the expression of all the evaluated genes thus establishing the link between YAP target genes expression and *GLO1* status in cancer cells. Efficient YAP silencing in *GLO1*-depleted MDA-MB-231 is shown in *Figure 5—figure supplement 1A*. This result led us to search for a statistical association between YAP and *GLO1* expression levels using a gene expression dataset of 103 primary mammary tumors (*Iwamoto et al., 2011*). However, global YAP expression did not show any significant correlation with *GLO1*. We reasoned that YAP activity, rather than its total expression level, would better reflect YAP nuclear accumulation related to MG stress. Indeed, we found that the expression of YAP target genes and *GLO1* showed a significant inverse correlation in breast cancer patients. Top 12 genes that displayed the highest Rp Pearson correlation coefficient are reported in *Figure 5—figure supplement 1B*. These data indicate that high carbonyl stress driven by low *GLO1* expression in human malignant mammary tumors is significantly associated with an elevated YAP activity.

Data gathered so far indicate that MG favors YAP activity in cancer cells. In order to assess more concretely the impact of MG stress on breast cancer cells through YAP activation, we next focused on CTGF gene expression, a well-described YAP transcriptional target (*Zhao et al., 2008*) that has been linked to YAP pro-growth and tumorigenic functions. We performed chromatin IP assays to assess the presence of YAP at CTGF promoter in both sparse and confluent MDA-MB-231 cells. In low-density cultured cells, YAP was bound to CTGF promoter, whereas in confluent cells, YAP was not detectable which is consistent with YAP absence in the nucleus of high-density cells. In contrast, YAP was found at CTGF promoter in MG-treated confluent cells at a comparable level to that in sparse cells (*Figure 5B*). Immunoblot of YAP IP loading is shown in *Figure 5—figure supplement 2A* CTGF mRNA level was not significantly affected by MG treatment in MDA-MB-231 cells (*Figure 5C*). Smad2/3 collaborate with TEAD and YAP to form an active transcriptional complex at CTGF promoter in breast cancer cells (*Hiemer et al., 2014*; *Fujii et al., 2012*). MDA-MB-231 cells cultured in presence of MG and treated with TGF-β (2.5 ng/ml) responded by a two-fold increase of CTGF mRNA level compared to TGF-β alone confirming the requirement of TGF-β pathway activation for MG-mediated induction of CTGF expression (*Figure 5C*). Smad2 and Smad3 phosphorylation following TGF-β treatment in MDA-MB-231 cells was not affected by MG treatment (*Figure 5—figure supplement 2B*) indicating that CTGF up-regulation was linked to YAP accumulation in

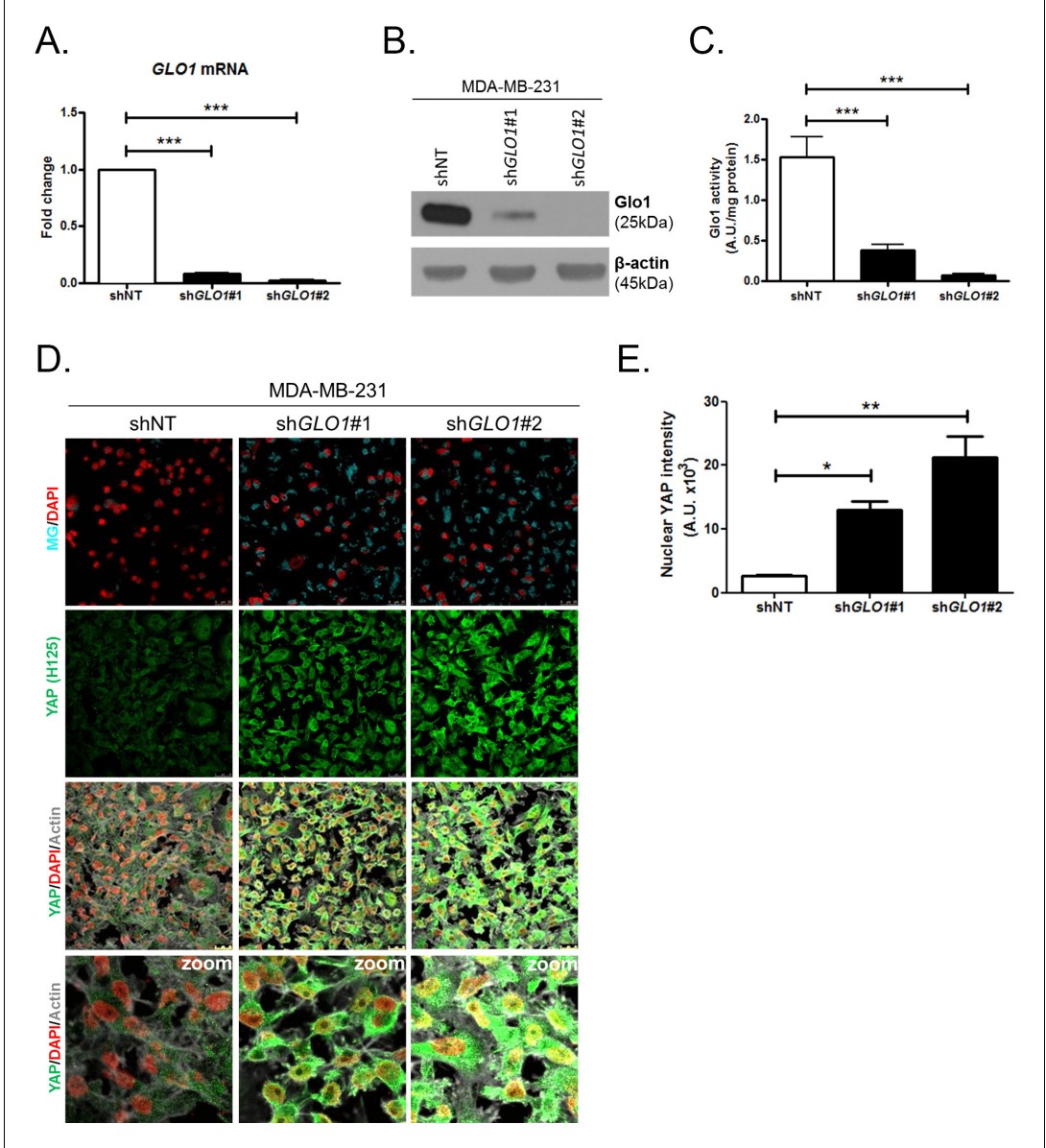

**Figure 4.** YAP cellular accumulation in sh*GLO1* MDA-MB-231 clones. (**A**) *GLO1* mRNA, (**B**) protein and (**C**) activity level in MDA-MB-231 shNT control and sh*GLO1*#1 and #2. (**D**) YAP immunofluorescence (Santa Cruz antibody, H125) in MDA-MB-231 silenced for *GLO1* (sh*GLO1*#1 and #2) cultured from low to high density. Detection of MG was performed using MBo-specific fluorescent probe. Data are representative of three independent experiments. Magnification 630x. Zoomed pictures are shown when indicated. (**E**) Quantification of nuclear YAP corresponding to D experiment. All data were analyzed using one-way ANOVA followed by Dunnett post-test and shown as the mean values ± SEM of at least three independent experiments. *p<0.05, **p<0.01 and ***p<0.001.

The following figure supplement is available for figure 4:

**Figure supplement 1.** Carnosine and aminoguanidine MG scavengers reverse YAP accumulation in *GLO1*-depleted MDA-MB-231.

presence of active TGF-β pathway effectors. In agreement with this deduction, we showed that YAP silencing prevented MG-mediated CTGF mRNA induction in presence of TGF-β in confluent MDA-MB-231 cells (*Figure 5D*). Efficient YAP silencing was shown at the mRNA (*Figure 5—figure supplement 2C*) and protein (*Figure 5—figure supplement 2D*) levels.

Association with TEAD transcription factors is essential in mediating YAP-dependent gene expression (*Zhao et al., 2008*). As shown in *Figure 5E* and *Figure 5—figure supplement 2E*, YAP

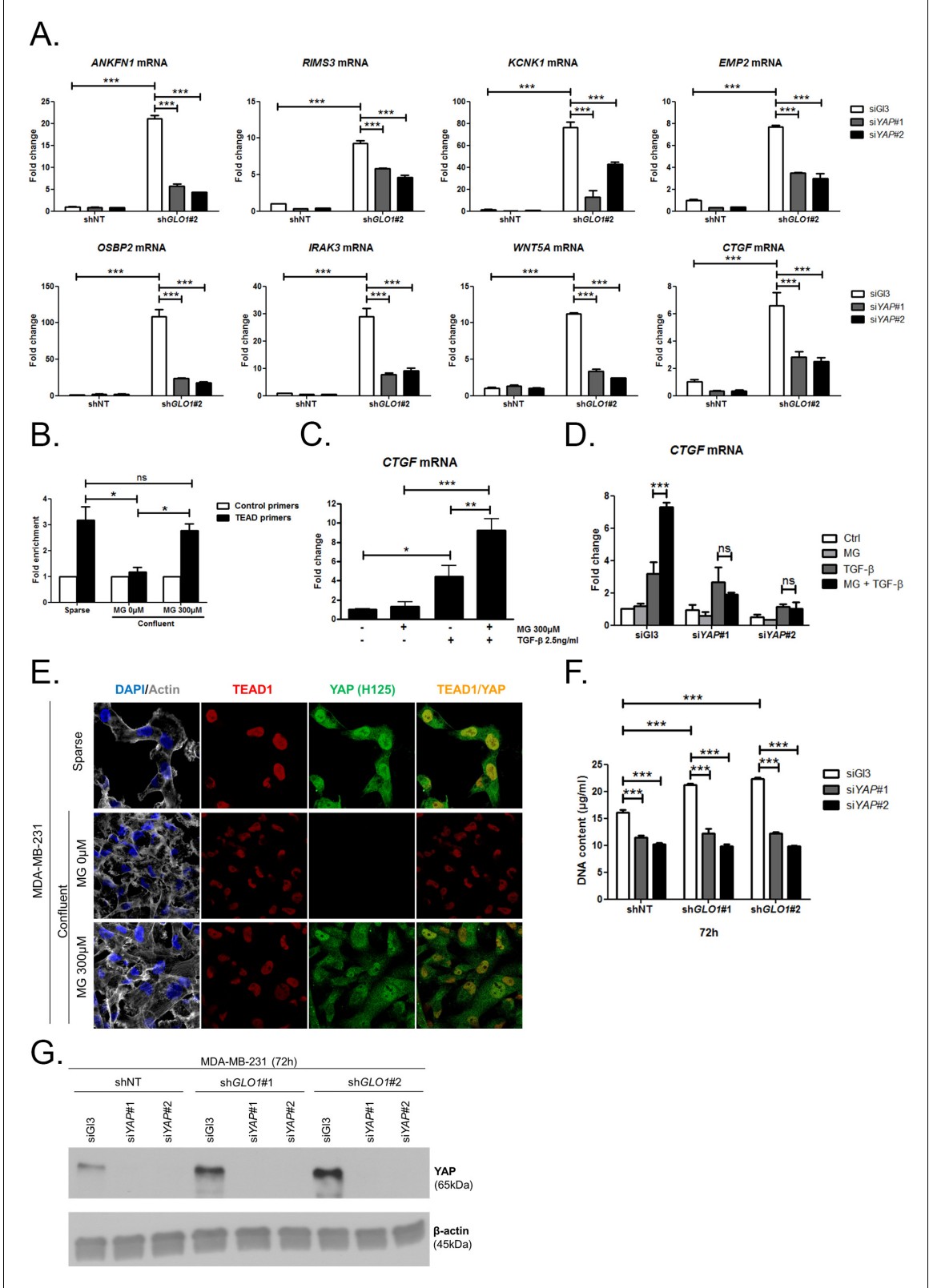

**Figure 5.** MG induces YAP co-transcriptional activity in breast cancer cells. (**A**) Stable knockdown of *GLO1* (sh*GLO1#2*) in MDA-MB-231 results in upregulation of several YAP target genes (ANKFN1, RIMS3, KCNK1, EMP2, OSBP2, IRAK3, WTN5A and CTGF) at the mRNA level as assessed by qRT-PCR. Silencing of YAP using two independent siRNAs (siYAP#1 and #2, 48 hr post-transfection) significantly reversed YAP target genes induction in *GLO1* depleted cells. Data were analyzed using two-way ANOVA followed by Bonferroni post-test and shown as the mean values ± SD of one

*Figure 5 continued on next page*

*Figure 5 continued*

representative experiment (n = 4). (**B**) Chromatin immunoprecipitation of YAP at the CTGF promoter in sparse and confluent MDA-MB-231 cells treated or not with MG. TEAD PCR primers, and not control primers, target TEAD binding site on CTGF promoter (see sequences under 'Materials and methods' section). The use of TEAD primers indicated that YAP was present at the CTGF promoter in sparse cells (positive control) and in confluent MG-treated cells when compared to untreated confluent cells. Data were analyzed using one-way ANOVA followed by Newman-Keuls post-test and shown as the mean values ± SEM of three independent experiments. (**C**) CTGF mRNA level assessed by qRT-PCR in MDA-MB-231 cells treated with MG 300 μM until confluence and then with TGFβ 2.5 ng/ml during 2 hr. Data were analyzed using two-way ANOVA followed by Bonferroni post-test and shown as the mean values ± SEM of five independent experiments. (**D**) MG-mediated CTGF induction in presence of TGFβ is not observed upon YAP silencing (siYAP#1 and #2) when compared to control (siGl3) cells. Data were analyzed using two-way ANOVA followed by Bonferroni post-test and shown as the mean values ± SEM of three independent experiments. (**E**) YAP (Santa Cruz antibody, H125) and TEAD1 IF co-localization in MDA-MB-231 cells cultured under low (Sparse) density used as positive control and in high-density cultured cells (Confluent) in presence of MG. Magnification 630x. Data are representative of three independent experiments. (**F**) DNA quantification assay showing an increased proliferation of *GLO1*-silenced MDA-MB-231 (sh*GLO1*#1 and #2) compared to control (shNT) at 72 hr. Silencing of YAP (siYAP#1 and #2) reversed this effect. Data were analyzed using two-way ANOVA followed by Bonferroni post-test and shown as the mean values ± SEM of four independent experiments. (**G**) Validation of YAP silencing by Western blot in MDA-MB-231 sh*GLO1* cells after 72 hr related to panel F and ***Figure 5—figure supplement 2F***. *p<0.05, **p<0.01, ***p<0.001 and ns = not significant.

The following figure supplements are available for figure 5:

**Figure supplement 1.** Inverse correlation between *GLO1* and YAP target genes expression.

**Figure supplement 2.** MG induces YAP co-transcriptional activity in breast cancer cells.

**Figure supplement 3.** MG increases YAP-mediated migratory potential in breast cancer cells.

and TEAD1 co-localized in MG-treated confluent MDA-MB-231 cells, and in sparse cultured cells used as positive control, whereas this co-localization was not detected in untreated confluent cells. YAP activation stimulates cancer cell growth. We next challenged cell proliferation induced by MG stress in *GLO1* depleted MDA-MB-231 cells. The time course of *GLO1*-depleted cells proliferation showed a marked difference compared to control cells (***Figure 5—figure supplement 2F***). At 72 hr, this increase in cell proliferation was significantly abrogated upon YAP silencing indicating that it is required to sustain MG-induced pro-growth effect (***Figure 5F***). Western blotting demonstrated both YAP increase upon *GLO1* silencing and efficient YAP silencing (***Figure 5G***).

YAP/TAZ have been previously reported to promote EMT in human breast epithelial cells (***Lei et al., 2008***; ***Overholtzer et al., 2006***). To test whether MG-induced YAP activation launched EMT in breast cancer cells, immunoblotting was performed to examine the expression of well-characterized EMT markers. We observed an increase of vimentin and a decrease of E-cadherin expression, attesting of an EMT process in MDA-MB-468 cells treated with MG (***Figure 5—figure supplement 3A***). Next, we observed by immunofluorescence that E-cadherin network was disrupted in MG-treated cells (***Figure 5—figure supplement 3B***). Consistent with EMT induction, MG-treated MDA-MB-468 cells showed an enhanced migration potential that was efficiently reversed to basal level upon YAP silencing (***Figure 5—figure supplement 3C and D***).

## MG favors LATS1 kinase degradation through the proteasome in breast cancer cells

To gain insight into possible mechanisms by which MG regulates YAP activity, we first considered that YAP could be a direct target of MG glycation. However, MG-adducts immunoprecipitation in MDA-MB-231 treated with MG did not allow western blot detection of YAP (data not shown). LATS1/2 are the main upstream Hippo pathway kinases that phosphorylate YAP, thus preventing its nuclear translocation and oncogenic activity (***Figure 6A***). We hypothesized that MG-mediated sustained YAP nuclear localization could be related to a relaxed LATS1/2 control notably due to its decreased expression. We demonstrated by Western blotting that LATS1, and not LATS2, was significantly decreased upon MG treatment (300 and 500 μM) in both glycolytic MDA-MB-231 and MDA-MB-468 and non-glycolytic MCF7 breast cancer cells (***Figure 6B***). LATS1 mRNA levels were not affected by MG in the three breast cancer cell lines (***Figure 6—figure supplement 1A***). A previous study has shown that LATS1 kinase degradation occurs through polyubiquitination and the

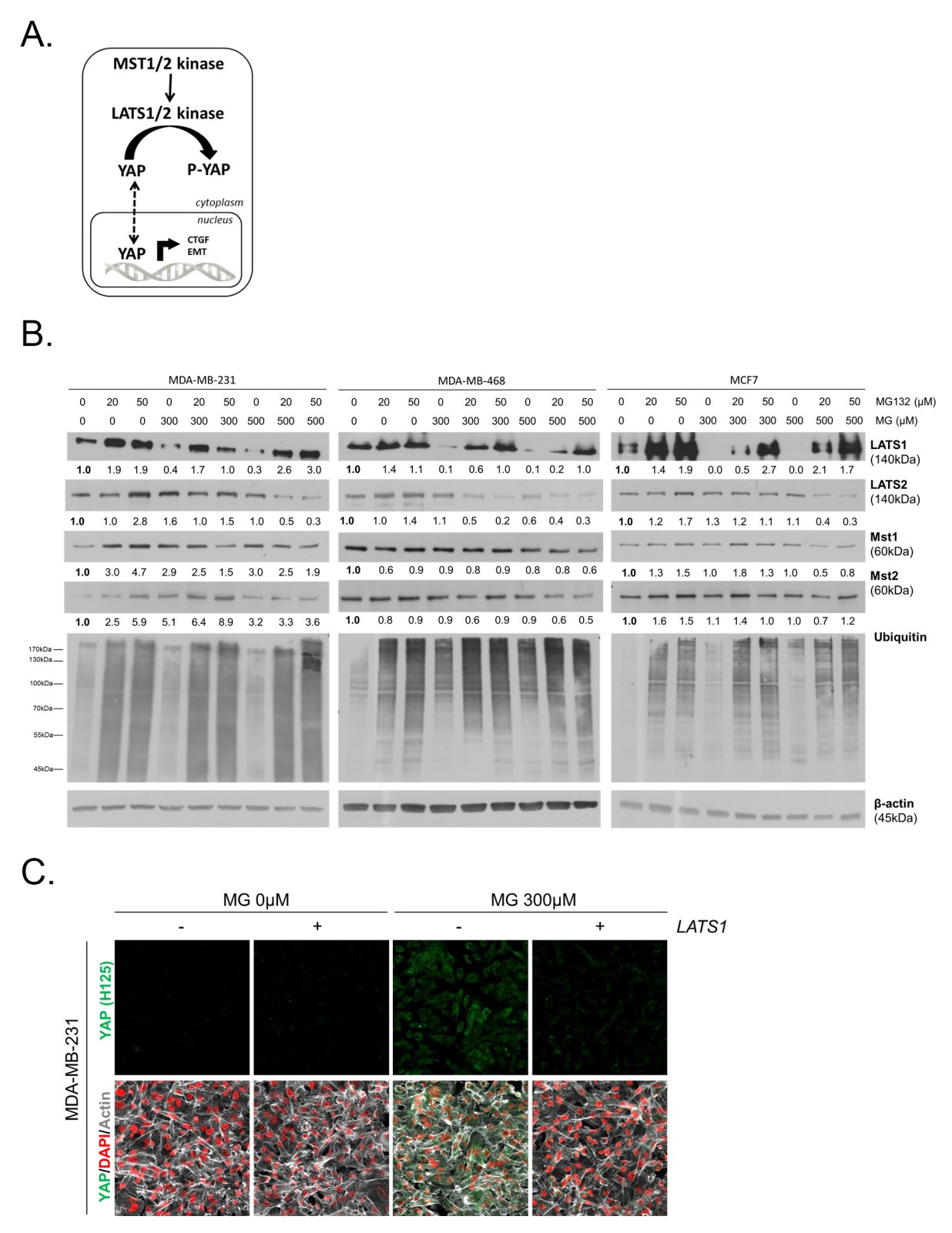

**Figure 6.** MG induces LATS1 kinase decrease in breast cancer cells. (A) Schematic representation of the Hippo pathway focused on MST1/2 and LATS1/2 kinases control of nuclear-cytoplasmic shuttling of YAP co-transcription factor. (B) LATS1, LATS2, MST1 and MST2 expression in MDA-MB-231, MDA-MB468 and MCF7 cells treated with MG (300 and 500 µM) in presence of increasing concentrations of MG132 proteasome inhibitor during 6 hr using Western blot. Ubiquitin immunoblot were performed to validate proteasome inhibition by MG132. Immunoblot data were quantified by

*Figure 6 continued on next page*

*Figure 6 continued*

densitometric analysis and normalized for β-actin. Numbers represent fold increase relative to the condition shown with bold number. (**C**) YAP immunofluorescence (Santa Cruz antibody, H125) in MDA-MB-231 cells transiently transfected with LATS1 expression vector (+) or empty vector used as control (-) and then treated with MG (300 µM) until confluence. All data are representative of three independent experiments.

The following figure supplement is available for figure 6:

**Figure supplement 1.** MG leads to YAP cellular accumulation through LATS1 expression decrease.

proteasome pathway in breast cancer cells (*He et al., 2016*). In good accordance, the treatment of breast cancer cells with MG132 proteasome inhibitor induced LATS1 increase (*Figure 6B*). Next, we verified whether MG favored LATS1 decrease through proteasome degradation. As shown in *Figure 6B*, the use of MG132 proteasome inhibitor reverted MG-induced LATS1 decrease. Next, we explored whether LATS1 decrease could explain, at least in part, the sustained YAP nuclear localization induced by MG. Accordingly, when we overexpressed LATS1, we were able to revert MG effects on YAP accumulation as assessed by immunofluorescence using two anti-YAP antibodies in MDA-MB-231 cells (*Figure 6C* and *Figure 6—figure supplement 1B*), and in the other breast cancer cell lines analyzed (*Figure 6—figure supplement 1C and D*). LATS1 overexpression is shown using anti-Flag and anti-LATS1 antibodies (*Figure 6—figure supplement 1E*). Data gathered so far indicate that MG decreases LATS1 expression in breast cancer cells, through the proteasome, which leads to sustained activity of YAP in the nucleus.

## MG induces post-translational glycation of Hsp90 and affects its chaperone activity on LATS1

To explore further MG mechanism of action on LATS1, we first excluded the possibility of (a) an interference of MG with the expression of Mst1/2 kinases directly upstream of LATS1 along the Hippo pathway (*Figure 6B*) and (b) a direct glycation of LATS1 by MG using immunoprecipitation technique (data not shown). Then, we got interested in LATS1 as a client of Hsp90 chaperone protein (*Huntoon et al., 2010*). Indeed, LATS1 kinase expression level and activity are dependent on its stabilization by Hsp90. 17-AAG, a potent Hsp90 inhibitor, disrupts LATS1 tumor suppressor activity in human cancer cells (*Huntoon et al., 2010*). *HSP90* mRNA level was not modulated by MG treatment in breast cancer cells (*Figure 7—figure supplement 1A*). Therefore, we sought to explore whether MG could modify Hsp90, thus indirectly impacting on LATS1 stability and degradation. The incubation of human recombinant Hsp90 with MG followed by MS analysis revealed the modification of several lysine and arginine residues notably yielding to the formation of carboxyethyllysine (CEL) and argpyrimidine/hydroimidazolone adducts, respectively (*Figure 7—source data 1*). Next, we examined whether endogenous MG-modified Hsp90 could be detected in MG-treated MDA-MB-231 cells. Immunoprecipitation of MG-treated MDA-MB-231 extracts using anti-argpyrimidine MG adducts and followed by Hsp90 immunoblot analysis allowed us to detect a basal level of glycated Hsp90 in MDA-MB-231 glycolytic cells that was further enhanced upon MG treatment (*Figure 7A*). Hsp27, which is recognized as a major MG target in cancer cells, was also efficiently detected in argpyrimidine immunoprecipitates (*Figure 7A*). Consistently, the reverse IP experiment using an anti-Hsp90 antibody allowed the detection of both argpyrimidine and hydroimidazolone MG-adducts of the expected molecular weight in MG treated cells (*Figure 7B*). Argpyrimidine immunoprecipitates subjected to MS analysis revealed the presence of modified Hsp90 on several residues (*Figure 7—source data 2*). Glycation hot spots observed on exogenous and/or endogenous Hsp90 are summarized in *Figure 7C* and detailed amino acid sequence is provided in *Figure 7—figure supplement 1B* The mapping of MG modifications on Hsp90 amino-acid sequence indicated that functionally important domains involved in both substrate/co-chaperone and ATP binding showed several glycated residues (*Figure 7C*) suggesting that Hsp90 activity could be affected. Recombinant human Hsp90 (rhHsp90) was effectively modified by MG and was protected by carnosine MG-scavenger as assessed using both anti-argpyrimidine and anti-hydroimidazolone antibodies in Western blot experiments (*Figure 7D*). Using an in vitro enzymatic assay, MG decreased rhHsp90 activity to an extent that was comparable to that seen using 17-AAG (*Figure 7E*). Incubation of rhHsp90 with MG in presence of carnosine efficiently reversed this effect indicating for the first time that direct MG

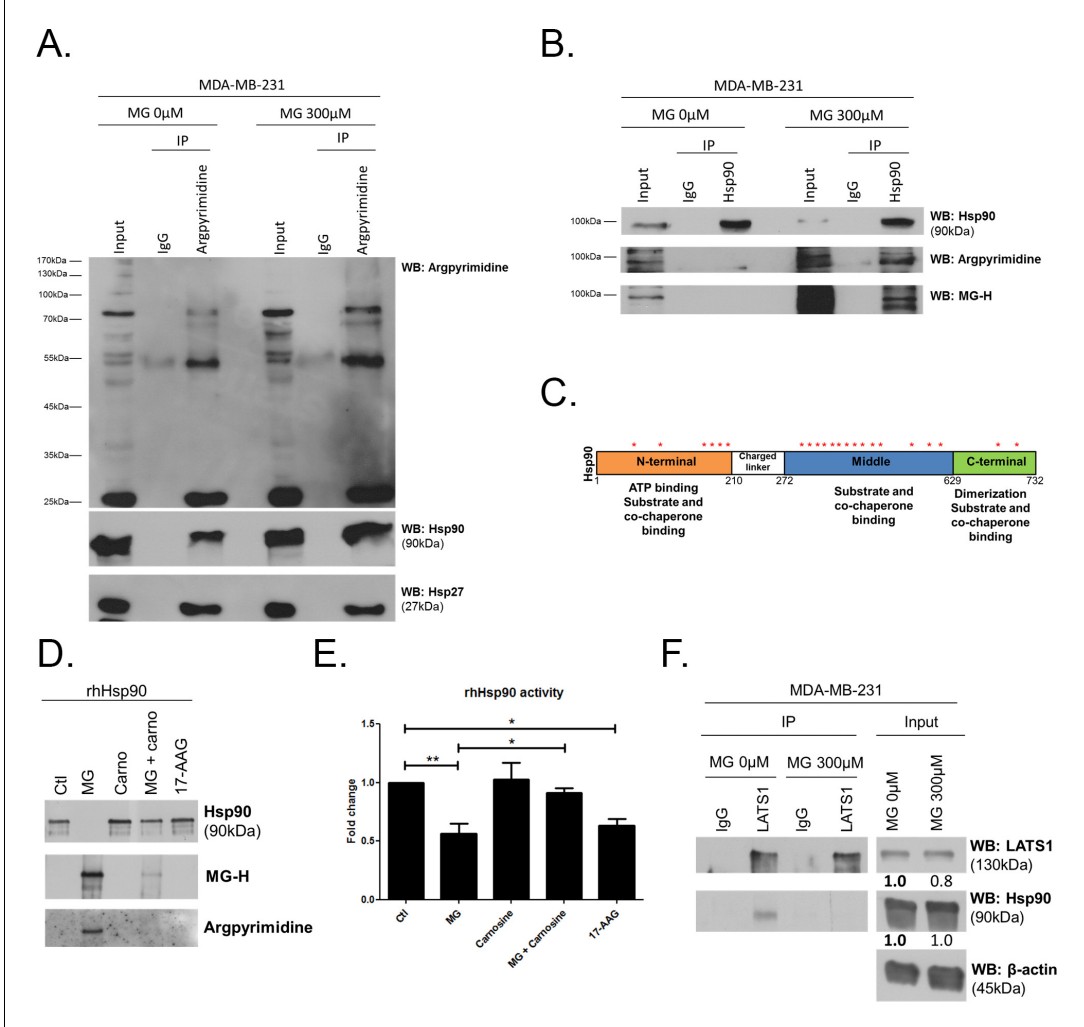

**Figure 7.** MG induces Hsp90 post-translational glycation in breast cancer cells. (**A**) Immunoprecipitation of MG adducts on MG-treated MDA-MB-231 cells (300 μM, 6 hr) using a specific anti-argpyrimidine monoclonal antibody. Mouse immunoglobulins (IgG) were used as control. Total cell lysates (Input) and immunoprecipitates (IP) were immunoblotted for argpyrimidine, Hsp90 and Hsp27. (**B**) Under the same conditions as in **A**, MDA-MB-231 cell lysates were immunoprecipitated using anti-Hsp90. Inputs and IPs were immunoblotted using Hsp90 antibody and two specific antibodies directed against MG-adducts (argpyrimidine and hydroimidazolone MG-H). (**C**) Schematic representation of Hsp90 protein domains where hot spots (*) of endogenously and/or exogenously MG-modified residues are indicated. See also detailed amino acid sequence in *Figure 7—figure supplement 1B*. (**D**) Western blot analysis using the indicated antibodies on recombinant human Hsp90 (rhHsp90) incubated in presence of MG ± carnosine (10 mM) or 17-AAG Hsp90 inhibitor (1 μM) during 24 hr. (**E**) Hsp90 ATPase activity was decreased after incubation with MG or 17-AAG. This effect is efficiently blocked in presence of carnosine MG scavenger. Data were analyzed using two-way ANOVA followed by Bonferroni post-test and shown as the mean values ± SEM of five independent experiments. *p<0.05 and **p<0.01. (**F**) Co-immunoprecipitation of LATS1 and Hsp90 from MDA-MB-231 cells treated with MG 300 μM during 24 hr reveals a decreased interaction between the two proteins. Immunoblot data were quantified by densitometric analysis and normalized for β-actin. Numbers represent fold increase relative to the condition shown with bold number. All data are representative of three independent experiments.

The following source data and figure supplement are available for figure 7:

**Source data 1.** MG modifications on human recombinant Hsp90.
**Source data 2.** MG modifications on endogenous Hsp90.
**Figure supplement 1.** MG induces Hsp90 post-translational glycation.

glycation of Hsp90 affects its ATPase activity (*Figure 7E*). Furthermore, both MG and 17-AAG decreased LATS1 expression in the three breast cancer cell lines under study (*Figure 7—figure supplement 1C*). Next, we further documented LATS1 binding to Hsp90 in the context of MG treatment. LATS1 immunoprecipitates contained detectable Hsp90 however this interaction was disrupted in presence of MG in MDA-MB-231 cells (*Figure 7F*). Collectively, our findings show that MG relieves LATS1 control on YAP nuclear localization through a mechanism identifying for the first time MG-mediated post-translational glycation and inactivation of Hsp90 in cancer cells.

### *GLO1*-depleted breast cancer cells show an increased tumorigenic and metastatic potential in a mouse xenograft model

Data gathered so far indicate that MG stress favors sustained YAP pro-proliferative and pro-migratory activity in breast cancer cells. Next, we explored the biological relevance of this observation for tumor growth and metastases development. Stably *GLO1*-depleted MDA-MB-231 cells that were grafted subcutaneously in mice showed an increased tumor weight and volume that reached significance for sh*GLO1*#2-silenced clones (*Figure 8A*). Further exploration of sh*GLO1* experimental tumors using immunoblotting revealed the effective in vivo induction of argpyrimidine adducts and a strong inverse relationship between *GLO1* silencing and total YAP expression (*Figure 8B and C*). In *GLO1*-silenced experimental tumors, we further demonstrated a specific increase of YAP in the nucleus of tumor cells using immunohistochemistry (*Figure 8D and E*). Elevated proportion of Ki67 positive cells in sh*GLO1* tumors sustained the observed increased tumor growth, as shown and scored in *Figure 8D and E*, respectively. In order to explore further the association between high MG, YAP activity and tumor growth, we used the in vivo chicken chorioallantoic membrane assay (CAM). Grafted *GLO1*-depleted cells on the CAM showed increased growth as assessed by the measure of tumor volume and compared to control cells (*Figure 8F and G*). Remarkably, YAP knockdown with two independent siRNAs further demonstrated the causative role of YAP in MG-induced tumor growth. Indeed, YAP silencing in sh*GLO1* cells efficiently reverted tumor growth to control levels (*Figure 8F and G*). As shown on mice experimental tumors, we observed significant YAP nuclear localization in CAM sh*GLO1* tumors (*Figure 8H and I*). YAP silencing was maintained for the entire duration of the CAM assay experiment (*Figure 8—figure supplement 1*).

Following the assessment of *GLO1*-silencing impact on tumor growth, we next evaluated the metastatic behavior of *GLO1*-depleted breast cancer cells. After surgical removal of the primary tumors, the mice were followed for metastases development during an additional period of 6 weeks. The follow-up of the mice showed that lung metastases were detectable already after 3 weeks in *GLO1*-depleted conditions but not in control condition. After 6 weeks post-tumor removal, metastasized tumors were observed in the lungs of *GLO1*-depleted mice (68%) when compared to control (20%). To evaluate further lung colonization, we performed human vimentin immunohistochemical detection that revealed a significant increase of both number and size of metastatic foci in *GLO1*-depleted condition (*Figure 9A*), as quantified in *Figure 9B*. Using serial sections, we assessed efficient *GLO1* depletion on the same metastatic foci (*Figure 9A*). These data demonstrate that breast cancer cells undergoing a carbonyl stress show enhanced growth and metastatic capacity thus highlighting an unexpected pro-tumoral role for MG endogenous accumulation. Finally, to better assess the importance of MG stress on metastatic dissemination, sh*GLO1* mice received carnosine (10 mM) in drinking water from the day of primary tumor removal until the end of the experiment (during 6 weeks). We observed a significant decrease of lung colonization in sh*GLO1* mice treated with carnosine when compared with control mice (*Figure 9C and D*). Collectively, our data indicate that the procancer effects of carbonyl stress unveiled here are tightly associated with YAP enhanced activity and can be efficiently blocked using a MG scavenger.

## Discussion

Cancer cell metabolism is characterized by an enhanced uptake and utilization of glucose through aerobic glycolysis. This overactive metabolism switch leads unavoidably to the formation of potent glycating agents such as MG. However, the concept of a causal relationship between non-enzymatic glycation and cancer progression is still in its early days. Here, we demonstrate that MG-mediated carbonyl stress interferes with LATS1 kinase, without affecting LATS2 and Mst1/2 kinases, to induce sustained YAP and TAZ nuclear localization. The hypermethylation of the promoter region of LATS1

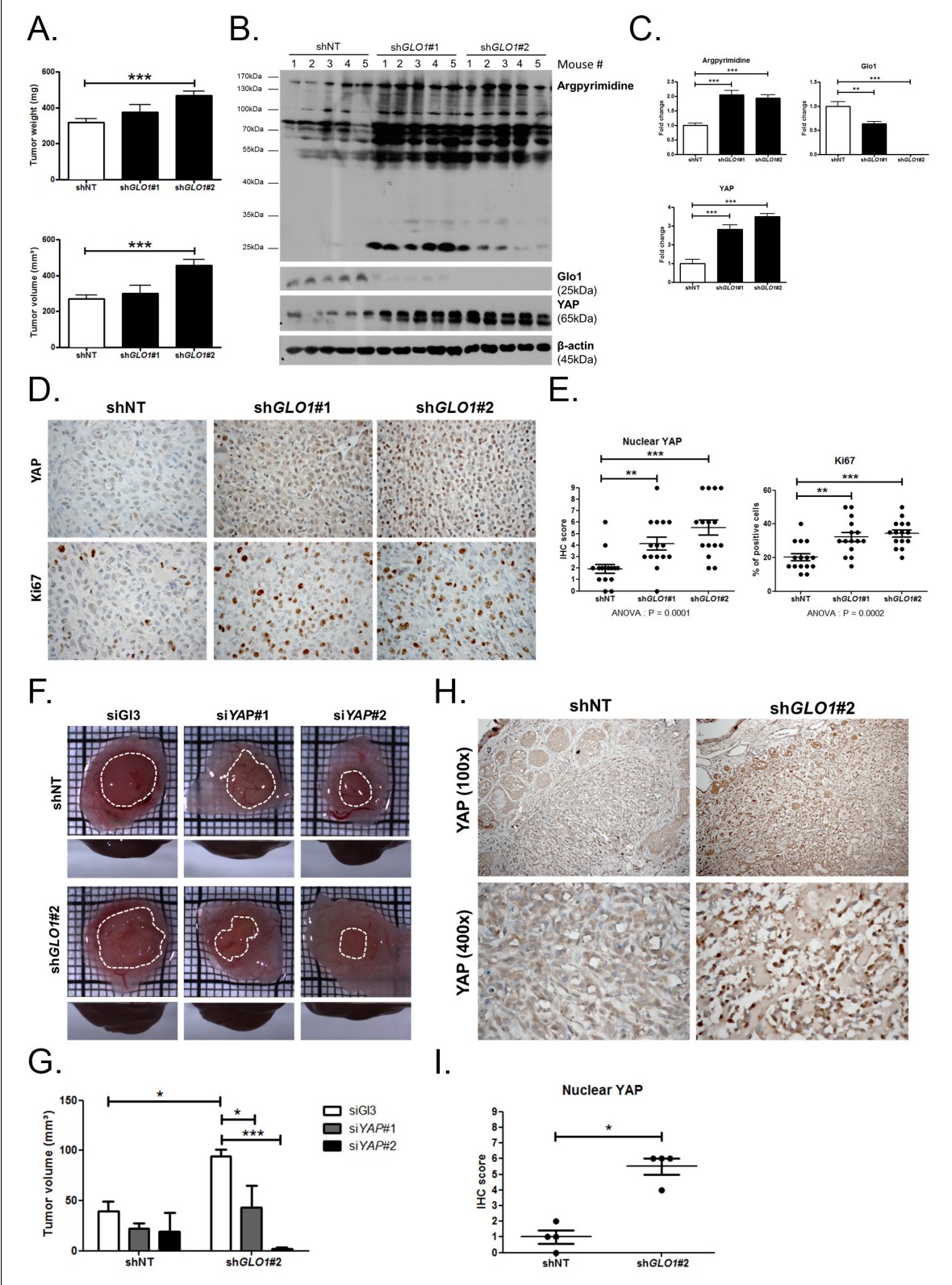

**Figure 8.** *GLO1*-depleted breast cancer cells show an increased tumorigenic potential in vivo. (**A**) MDA-MB-231 sh*GLO1*#1 and #2 and control shNT cells were injected subcutaneously in NOD-SCID mice (15 mice/group). After 4 weeks, primary tumors were surgically removed and weighed. Tumor weight (mg) and volume (mm³) were analyzed using one-way ANOVA followed by Dunnett post-test and shown as the mean values ± SEM. (**B**) Western blot detection of argpyrimidine, Glo1 and YAP in five representative experimental primary tumors. β-actin is used for normalization. (**C**) Quantification

*Figure 8 continued on next page*

*Figure 8 continued*

of the western blot shown in panel B. Data were analyzed using one-way ANOVA followed by Dunnett post-test and shown as the mean values ± SEM. (**D**) Representative YAP and Ki67 IHC staining in experimental primary tumors. (**E**) Quantification of IHC shown in panel D. Each dot represents one case and bars represent mean ± SEM. Data were analyzed using one-way ANOVA Kruskal-Wallis test followed by Dunn post-test (YAP) and one-way ANOVA followed by Dunnett post-test (Ki67). (**F**) *GLO1*-depleted MDA-MB-231 (sh*GLO1#2*) and control shNT cells were transfected with YAP siRNAs (siYAP#1 and 2) and grown on the chicken chorioallantoic membrane (CAM). After 7 days, tumors were collected and measured. Top and profile views of representative experimental CAM tumors are shown. (**G**) Tumor volumes (4 tumors/condition) were analyzed using two-way ANOVA followed by Bonferroni post-test and shown as the mean values ± SEM. (**H**) Representative YAP immunostaining on *GLO1*-depleted CAM experimental tumors. (**I**) Quantification of nuclear YAP IHC shown in panel F. Each dot represents one case and bars represent mean ± SEM. Data were analyzed using Mann Whitney t test. *p<0.05, **p<0.01 and ***p<0.001.

The following figure supplement is available for figure 8:

**Figure supplement 1.** *GLO1*-depleted breast cancer cells show an increased tumorigenic potential in vivo.

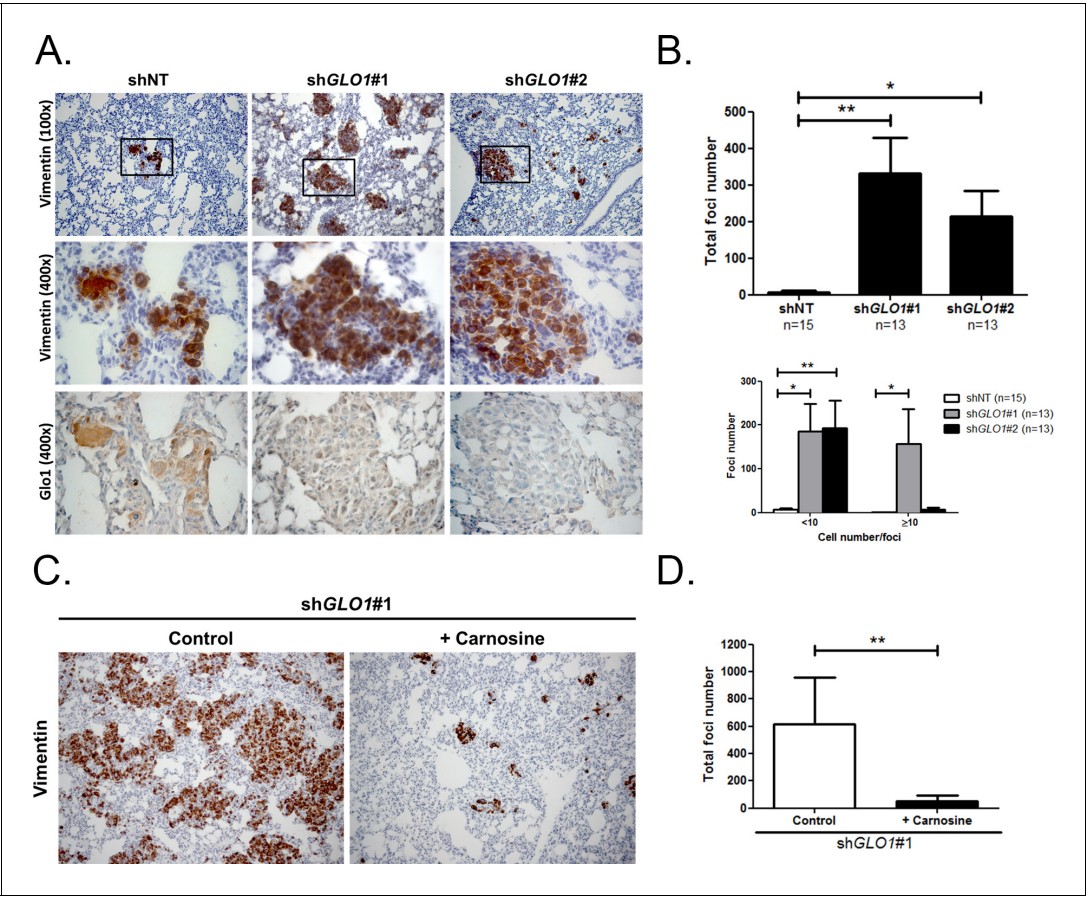

**Figure 9.** *GLO1*-depleted breast cancer cells show an increased metastatic potential in a mouse xenograft model. (**A**) MDA-MB-231 sh*GLO1#1* and #2 and control shNT cells were injected subcutaneously in NOD-SCID mice. After 4 weeks, primary tumors were surgically removed. Six weeks after tumor removal, mice were sacrificed and lungs were collected. We had to ethically sacrifice two mice in both sh*GLO1#1* and #2 groups before the end of the experiment. Representative human vimentin IHC highlights lung metastatic tumor lesions. Adjacent serial sections were used to perform Glo1 IHC staining. (**B**) Quantification of number and size of vimentin positive foci on whole lung sections. Data were analyzed using two-way ANOVA followed by Newman Keuls or Bonferroni post-test and shown as the mean values ± SEM. The number of mice per group is indicated on the graph. (**C**) MDA-MB-231 sh*GLO1#1* cells were injected subcutaneously in NOD-SCID mice (5 mice/group). After 4 weeks, primary tumors were surgically removed and mice were treated with carnosine (10 mM) in drinking water. Six weeks after tumor removal, mice were sacrificed and lungs were collected. Human vimentin IHC staining of whole lung sections highlights metastatic tumor lesions. Magnification 100x. (**D**) Quantification of vimentin-positive foci on whole lung sections. Data were analyzed using unpaired student's t test and shown as the mean values ± SEM. *p<0.05 and **p<0.01.

gene (50% of breast tumors) is associated with an aggressive breast cancer phenotype and poor survival (*Takahashi et al., 2005*). We show in this study that LATS1 proteasomal degradation is favored in presence of MG thus bringing to light a new concept according to which MG stress could directly and/or indirectly participate to the control of tumor suppressor genes in cancer cells without affecting their transcriptional rate.

YAP is regulated by diverse mechanisms including microenvironmental factors (cell crowding and ECM stiffness) and extracellular signaling (G-coupled receptors) (for review [*Moroishi et al., 2015*]). Our study meets a new trend of thoughts proposing that energy metabolism is an additional upstream signal that regulates YAP oncogenic activity. Three independent studies recently established a link between the Hippo-YAP pathway and cellular energy stress using a glucose deprivation strategy (*DeRan et al., 2014*; *Mo et al., 2015*; *Wang et al., 2015*). These studies are in agreement that under low-glucose condition an AMPK-LATS1 axis inhibits YAP activity. Enzo and collaborators (*Enzo et al., 2015*) have recently demonstrated that aerobic glycolysis impacts on YAP/TAZ transcriptional activity through a mechanism involving phosphofructokinase 1 binding to TEAD transcription factors. A recent metabolic profiling study using breast cancer progression cellular models reported the induction of several glycolytic enzymes upon constitutive activation of YAP/TAZ factors (*Mulvihill et al., 2014*). Therefore, it is tempting to speculate that MG could regulate key glycolytic enzymes expression in a YAP-dependent manner thus creating a mutual regulatory loop where glycolysis-induced MG stress favors YAP activity, which in turn activates glycolysis. Considering that glucose metabolism inevitably leads to MG formation, one might speculate that any signaling pathway favoring the Warburg effect, e.g the Wnt signaling (*Pate et al., 2014*), will ultimately feed MG carbonyl stress in cancer cells.

We show for the first time that Hsp90 is post-translationally glycated by MG. Although several post-translational modifications have been previously reported to affect Hsp90 stability and chaperone function, our study importantly uncovers that a natural metabolite derived from glycolysis is involved in regulating Hsp90 in cancer cells. Quantitative glycation studies will help to determine the proportion of Hsp90 molecules that are glycated by MG, the degree of glycation of specific residues per molecule and to what extent Hsp90 activity is likely to be inhibited in cancer cells. Decreasing Hsp90 client binding has been considered as an attractive anti-cancer therapy because of its role in stabilizing the active form of a wide range of client oncoproteins and several synthetic Hsp90 inhibitors are now in clinical trials (*Neckers and Workman, 2012*). Nevertheless, cumulative evidence tends to prove that Hsp90-directed therapy also induces pro-cancer effects. For example, Hsp90 inhibition promotes prostate cancer growth through Src kinase activation (*Yano et al., 2008*) and favors breast cancer bone metastases formation (*Price et al., 2005*). It is generally estimated that over sixty percent of all cancers are glycolytic. This study showing that a glycolysis metabolite interferes with Hsp90 activity even more crucially raises significant concerns about the use of Hsp90 inhibitors as cancer treatment. The above-mentioned tumor-promoting effects related to Hsp90 inhibition could be potentially recapitulated under MG-mediated carbonyl stress condition in cancer cells. Consistent with this hypothesis and our data, LATS1 signalization in the Hippo pathway is rendered ineffective in ovarian cancer xenograft tumors from mice treated with an Hsp90 inhibitor (*Huntoon et al., 2010*).

MG is a potent cytotoxic compound and was first viewed as a potential therapeutic agent in cancer (*Kang et al., 1996*). However, the recent identification of MG-modified proteins with pro-tumorigenic potential indicated that MG could also support tumor progression. Three independent groups have shown that Hsp27 heat-shock protein is switched from a pro-apoptotic to an anti-apoptotic factor upon MG glycation and facilitates cancer cell evasion from caspase-dependent cell apoptosis (*van Heijst et al., 2006*; *Oya-Ito et al., 2011*; *Sakamoto et al., 2002*). These data and ours point to an important regulatory role of MG stress on Hsps which have been particularly shown to be overexpressed in a wide range of tumors and are associated with a poor prognosis and resistance to therapy (*Calderwood et al., 2006*).

*GLO1* appears to be a dual mediator for growth regulation in cancer as it has been described both as an oncogene and a tumor suppressor. On one hand, the search for copy number changes on a large set of cancer cell lines revealed that *GLO1* is amplified in many types of human cancer with breast tumors (22%) and sarcomas (17%) showing the highest rates (*Santarius et al., 2010*). Most of the recent studies aimed at the inhibition of *GLO1* to induce a toxic MG accumulation effectively showed a decreased tumor growth. These studies generally depicted *GLO1* as an amplified

and/or overexpressed oncogene and as a bad prognosis marker in different types of malignant tumors (*Zhang et al., 2014*; *Cheng, 2012*; *Hosoda et al., 2015*; *Antognelli et al., 2013*; *Fonseca-Sánchez et al., 2012*). On the other hand, a study aimed at functionally identifying tumor suppressor genes in liver cancer identified and validated *GLO1* as a tumor suppressor gene which knockdown using shRNAs increased tumor growth in a mouse model (*Zender et al., 2008*). Using stably depleted *GLO1* xenografts in vivo, we have also demonstrated the pro-tumorigenic and pro-metastatic role of endogenous MG accumulation in breast cancer cells. These results are consistent with (a) enhanced nuclear YAP and increased Ki67-positive proliferating cells in vivo, (b) increased YAP oncogenic activity including the induction of growth factors expression such as CTGF and the initiation of EMT process observed in vitro and (c) the positive correlation between high MG-adducts detection and nuclear YAP in human primary mammary tumors, thus supporting the clinical relevance of our findings.

Therefore, it can be expected that different cancer types, with different backgrounds and for instance different MG detoxification rates would react differently to MG stress. Remarkably, cell lines with *GLO1* amplification (*Santarius et al., 2010*) or high Glo1 expression (*Sakamoto et al., 2001*) are reported to be much more sensitive to Glo1 inhibitors such as BBGC than those without. Thus, suggesting that a high Glo1 activity, putatively associated with high MG production, is necessary for their survival. In our hands, *GLO1* knockdown or inhibition using BBGC did not induce any significant cell apoptosis and hence conferred pro-growth and pro-metastatic advantages to breast cancer cells. These apparently controversial results could be potentially ascribed to differences in the cell lines and animal models used. No doubt that the validation of *GLO1* as a target for cancer therapy will need a better characterization of those breast tumors that are more likely to be sensitive.

Many chemotherapeutic drugs used to treat cancer have been shown to exert their biologic activity through induction of oxidative stress. However, compelling experimental and clinical evidence indicates that this latter is diverted by cancer cells to promote their growth and resistance to apoptosis. Such promoting effect of carbonyl stress in cancer is inferred for the first time from our data and certainly awaits more comprehensive studies before confirmation. In a non-tumoral context, MG dual effect has been shown recently for neurons where MG is favorable to neurons development and activity while high MG levels are toxic (*Radu et al., 2012*).

Elevated blood concentrations of MG have been reported in type 2 diabetic patients (*Nakayama et al., 2008*) and plasma MG-derived hydroimidazolone was higher in type 1 diabetics as compared with non diabetics (*Han et al., 2009*). Future studies will have to assess the potential use of circulating MG and/or specific MG adducts as cancer biomarkers. Recent studies indicated that cancer in diabetic patients presents with a higher incidence and a poorer prognosis than in non-diabetic persons (*Ryu et al., 2014*; *Xu et al., 2014*). Our study hereby provides with a potential molecular mechanism for cancer-diabetes connection. A better understanding of MG pro-cancer effects could lead to the development of preventive and therapeutic strategies based on the scavenging of MG. Interestingly, both MG scavengers metformin and carnosine have been shown to exert anti-tumoral effects (*Shen et al., 2014*; *Giovannucci et al., 2010*) and metformin proved efficient to reduce systemic MG levels in diabetic patients (*Beisswenger et al., 1999*). DeRan and collaborators (*DeRan et al., 2014*) reported that metformin, and its more potent analog phenformin, inhibited YAP activity through AMPK signaling. Metformin is better known as a mitochondrial complex I inhibitor and a potent AMPK inducer, and it is somehow overlooked for its MG scavenging capacities. Accordingly, it is not excluded that metformin could have also exerted, at least in part, its inhibitory activity on YAP function through its MG scavenging properties. In support of this hypothesis, we have shown that high MG stress-tumor xenografts showed a significantly lower propensity to metastasize in animals supplemented with carnosine in their drinking water. The crucial role of glucose metabolism in aggressive tumors has logically directed cancer therapy research towards the use of anti-diabetic drugs as effective anti-cancer agents and metformin is actually tested in several anti-cancer clinical trials (*Quinn et al., 2013*).

Studies using mass spectrometry and antibodies directed against MG specific AGEs are currently underway in our laboratory in order to identify other targets of MG that will hopefully contribute to bring to light the critical position of MG-mediated carbonyl stress in cancer.

# Materials and methods

## Cell culture and reagents

Human breast cancer cell lines MDA-MB-231 and MCF7 were obtained from the American Type Culture Collection (ATCC, Rockville, MD, USA). Human breast cancer cell line MDA-MB-468 was kindly provided by Dr. C. Gilles (Laboratory of Tumor and Development Biology, University of Liège, Belgium). All cell lines were authenticated by STR profiling at the Leibniz-Institute DSMZ (Braunschweig, Germany) and were regularly checked for mycoplasma contamination using MycoAlert Mycoplasma Detection Kit (Lonza, Basel, Switzerland). Cells were either cultured in high-glucose DMEM (standard glucose concentration of 4.5 g/L, Lonza) or in DMEM medium with a glucose concentration of 1 g/l (low-glucose medium) both containing 10% fetal bovine serum (FBS, ThermoFisher Scientific, Waltham, MA, USA) and 2 mM L-glutamine (Lonza). One g/l glucose level is physiological and reflects the in vivo concentrations in human serum. Conversely, the routine culture media concentration of 4.5 g/l (high-glucose medium) corresponds to a diabetic condition. In our experimental model, sparse cells are defined as low-glucose cultured cells seeded at low density (for example, for a 6-well plate: MDA-MB-231, MCF7 and MDA-MB-468 were seeded at $2 \times 10^5$ cells/well). Confluent cells were obtained by seeding the same number of cells as indicated above and cultured for 3 days until they reached confluence. During these 3 days, cells were treated where indicated with methyl-glyoxal (MG, cat#M0252, Sigma-Aldrich, Saint-Louis, MO, USA) at micromolar concentrations every day. We excluded the presence of significant formaldehyde contamination (<3%) in MG (lot: BCBF6939V) by NMR analysis. The natural anti-glycation dipeptide L-carnosine (C9625), the MG scavenger aminoguanidine (396494), the proteasome inhibitor MG132 (C2211), the Hsp90 inhibitor 17-AAG (A8476) and the glycolysis inhibitor 2-deoxyglucose (2-DG, D8375) were obtained from Sigma-Aldrich. TGF-β was obtained from Roche (Mannheim, Germany. Human recombinant Hsp90α (rhHsp90) was obtained from Enzo Life Sciences (Farmingdale, NY, USA). S-p-bromobenzylgluta-thione cyclopentyl diester (BBGC), a cell-permeable inhibitor of Glo1, was synthesized as previously described (*Thornalley et al., 1996*). Anti-argpyrimidine antibody (mAb6B) specificity has been previously confirmed by competitive ELISA and it has been shown to not react with other MG-arginine adducts such as 5-hydro-5-methylimidazolone and tetrahydropyrimidine (*Oya et al., 1999*).

## Clinical tumor samples

Human breast tumor tissues (n = 87) were obtained from the Pathology Department of the University Hospital of Liège in agreement with ethical guidelines of the University of Liège (Belgium).

## Immunohistochemistry (IHC)

Formalin-fixed paraffin embedded sections were dewaxed and rehydrated. Sections were treated with 3% hydrogen peroxide in methanol for 30 min to block endogenous peroxidase activity. Antigen retrieval was performed in 10 mM sodium citrate buffer pH6 at 95°C for 40 min. To block non-specific binding sites, tissues were incubated with 1.5% normal serum (Vector Laboratories, Burlingame, CA, USA) for 30 min. Next, they were incubated with mouse anti-Argpyrimidine (mAb6B, 1:2000), rabbit anti-YAP (Santa Cruz (Dallas, TX, USA), H125, 1:100), mouse anti-Ki67 (Dako, Agilent Technologies, Santa Clara, CA, USA, 1:100), mouse anti-Glo1 (BioMac (Leipzig, Germany), 1:100) and mouse anti-vimentin (Ventana, Roche, 1:4) antibodies overnight at 4°C followed by incubation with an anti-mouse or anti-rabbit biotinylated secondary antibody (Vector Laboratories) for 30 min at room temperature (RT). Sections were then stained with avidin-biotin-peroxidase complex (Vectastain ABC Kit, Vector Laboratories) for 30 min followed by staining with 3,3' diaminobenzidine tetrachloride (DAB). Slides were finally counterstained with hematoxylin, dehydrated and mounted with DPX (Sigma-Aldrich). Tissue sections incubated without primary antibody showed no detectable immunoreactivity.

## Evaluation of immunohistochemical staining

The immunostaining was reviewed and scored by two examiners including an anatomopathologist (E.B). As we previously described (*Waltregny et al., 1998*), scoring of the staining was done according to the intensity of the staining (0, 1+, 2+, 3+) and the percentage of positive cancer cells within the tumor (0–25%, 25–50%, 50–75%, 75–100%). The results obtained with the two scales were

multiplied together, yielding a single score with steps of 0, 1, 2, 3, 4, 6 and 9. For argpyrimidine staining, scores of 0 to 2 were considered as low to intermediate staining (low/intermediate carbonyl stress) and scores from 3 to 9 were considered as high staining (high carbonyl stress). Expression status of YAP in tumor cells was assessed using the same scoring as described above according to YAP cellular compartment (nucleus and cytoplasmic). Ki67 immunostaining was evaluated as the percentage of nucleus positive cells present in experimental tumor tissue sections. Human vimentin detection was used to quantify (a) the total number of positive foci and (b) the number of cells per foci categorized as follows: < 10 and ≥10 vimentin positive cells. Metastatic foci number and size were counted in one whole lung section per mice.

## Immunofluorescence (IF) and evaluation of nuclear YAP staining

MDA-MB-231, MDA-MB-468 and MCF7 cells were plated on coverslips in 24-well plates. Sparse cells were seeded at $4 \times 10^4$ cells/well. Confluent cells were obtained by seeding the same number of cells as indicated above and treated for 3 days with MG until they reached confluence. For YAP, TAZ and/or TEAD1 staining, cells were fixed with 3% paraformaldehyde (PAF) for 20 min and then permeabilized with 1% Triton X-100. After blocking (3% BSA for 30 min), slides were incubated with rabbit anti-YAP (Santa Cruz, H125 or Cell Signaling, Danvers, MA, USA, 4912, 1:100), mouse anti-TAZ (BD Biosciences  Franklin, Lakes, NJ, USA), 1:50 (MCF7 cells) – 1:100 (MDA-MB-231 and MDA-MB-468 cells) and/or mouse anti-TEAD1/TEF1 (BD Biosciences, 1:100) antibodies diluted in 1% BSA overnight at 4°C. After washing with PBS, slides were incubated with anti-rabbit IgG AlexaFluor488, anti-mouse IgG AlexaFluor 488 or anti-mouse IgG AlexaFluor633 conjugated secondary antibodies (Life Technologies, Carlsbad, CA, USA), 1:1000) for 1 hr at RT. AlexaFluor568 Phalloidin (Invitrogen, Carlsbad, CA, USA) was used to stain actin filaments. For E-cadherin and YAP co-staining, cells were fixed in cold methanol for 10 min at −20°C. After rehydratation in PBS, slides were blocked in 3% BSA for 30 min and stained with mouse E-cadherin antibody (BD Biosciences, 1:200) and/or YAP antibody (Santa Cruz, H125, 1:100) diluted in 1% BSA for 1 hr at RT. After a washing step, slides were incubated with anti-mouse IgG AlexaFluor488 and/or anti-rabbit IgG AlexaFluor546 (Invitrogen). Nuclei were stained using DAPI (EMD Chemicals, San Diego, CA, USA). Coverslips were mounted on glass slides using Mowiol (Sigma-Aldrich) and observed using confocal microscope (Leica SP5). All microscope settings were kept the same for sparse and confluent cells imaging. To quantify the intensity of YAP nuclear staining, at least 5 pictures/condition (magnification 630x) were used. Using ImageJ software (RRID:SCR_003070) (*Schneider et al., 2012*) the intensity of YAP staining that colocalized with DAPI staining was measured. A mean of YAP staining intensity per nucleus was obtained. All the data are presented as the mean ± SEM of three independent experiments.

## Western blot

Cells were extracted in SDS 1% buffer containing protease and phosphatase inhibitors (Roche). Tissues samples were extracted in RIPA buffer (150 mM NaCl, 0.5% Na$^+$-deoxycholate, 1% Triton X-100, 0.1% SDS, 50 mM Tris-HCl pH7.5 and protease/phosphatase inhibitors). After incubation under rotation at 4°C during 30 min, tissues lysates were centrifuged at 14000 g for 15 min at 4°C. Protein concentrations were determined using the bicinchoninic acid assay (Pierce, Rockford, IL, USA). Twenty or 30 μg of proteins were separated by 7.5 to 12.5% SDS-PAGE and transferred to PVDF or nitrocellulose membranes. After blocking in TBS-Tween 0.1% containing 5% nonfat dried milk (Bio-Rad, Hercules, CA, USA), membranes were incubated with primary antibodies overnight at 4°C. Antibodies are listed in *Supplementary file 1*. Then, the membranes were exposed to appropriate secondary antibody at RT for 1 hr. The immunoreactive bands were visualized using ECL Western Blotting substrate (Pierce). Immunoblots were quantified by densitometric analysis and normalized for β-actin using ImageJ software. A representative western blot of three independent biological replicates is shown.

## siRNA transfection

YAP and *GLO1* small interfering RNAs (siRNA) and siGl3 irrelevant used as control were synthesized by Eurogentec (Liège, Belgium). Sequences are detailed in *Supplementary file 2*. Calcium phosphate-mediated transfections were performed using 20 nM of each siRNA.

## shRNA transfection

Lentiviral vectors (rLV) were generated with the help of the GIGA Viral Vectors platform (University of Liège). Briefly, Lenti-X 293T cells (Clontech, Montain View, CA, USA) were co-transfected with a pSPAX2 (a gift of Dr D. Trono, Addgene plasmid #12260) and a VSV-G encoding vector (*Emi et al., 1991*) along with a shRNA transfer lentiviral plasmid (*GLO1* shRNAs plasmids : Sigma-Aldrich, TRCN0000118627 (#1) and TRCN0000118631 (#2) or non-target (NT, anti-eGFP) shRNA plasmid (Sigma-Aldrich, SHC005)). Forty-eight, 72 and 96 hr post-transfection, viral supernatants were collected, filtrated and concentrated 100x by ultracentrifugation. The lentiviral vectors were then titrated with qPCR Lentivirus Titration Kit (ABM, Richmond, Canada). MDA-MB-231 cells were stably transduced with sh*GLO1*#1, sh*GLO1*#2 and shNT and selected with puromycin (0.5 µg/ml, Sigma-Aldrich).

## RNA isolation and quantitative reverse transcription-PCR (qRT-PCR)

RNA extraction was performed according to the manufacturer's protocol (NucleoSpin RNA, Macherey-Nagel, Duren, Germany). Reverse transcription was done using the Transcription First Strand cDNA Synthesis Kit (Roche). Hundred ng of cDNA were mixed with primers, probe (Universal ProbeLibrary System, Roche) and 2x FastStart Universal Probe Master Mix (Roche) or Fast Start SYBR Green Master Mix (Roche). Q-PCR were performed using the 7300 Real Time PCR System and the corresponding manufacturer's software (Applied Biosystems, Foster City, CA, USA). Relative gene expression was normalized to 18S rRNA. Primers were synthesized by IDT (Leuven, Belgium) and their sequences are detailed in *Supplementary file 3*. Three technical replicates of each sample have been performed and data are presented as mean ± SEM or ± SD (as indicated in figure legends) of minimum 3 biological replicates.

## Cellular MG quantification

MBo (Methyl diaminobenzene-BODIPY) was used to detect endogenous MG in different conditions. The cells were treated with 5 µM MBo in complete medium as previously described (*Wang et al., 2013*). After 1 hr, the cells were washed with PBS and incubated in low- or high-glucose medium for 6 (FACS) and 24 hr (confocal microscopy). Cells were either trypsinized and analyzed by flow cytometry (BD Biosciences FACSCanto), or fixed with PAF and prepared for confocal microscope visualization as described above.

## Nuclear magnetic resonance (NMR) analysis

Five hundred microliters of conditioned culture media (24 hr) were supplemented with 100 µl of deuterated phosphate buffer (pH7.4), 100 µl of a 35 mM solution of maleic acid and 10 µl of TMSP. The solution was distributed into 5 mm tubes for NMR measurement. $^1$H-NMR spectra were acquired using a 1D NOESY sequence with presaturation. The Noesypresat experiment used a RD-90°-t1-90°-tm-90°-acquire sequence with a relaxation delay of 4 s, a mixing time tm of 10msec and a fixed t1 delay of 4 µs. Water suppression pulse was placed during the relaxation delay (RD). The number of transient is 32 (64K data points) and a number of 4 dummy scans is chosen. Acquisition time is fixed to 3.2769001 s. Lactate dosages were achieved by integrations of the lactate signal at 1.34ppm using maleic acid as internal standard. Deuterium oxide (99.96% D) and trimethylsilyl-3-propionic acid-*d*4 (TMSP) were purchased from Eurisotop (St-Aubin, France), phosphate buffer powder 0.1M and maleic acid were purchased from Sigma-Aldrich. The NMR spectra were recorded at 298 K on a Bruker Avance spectrometer operating at 500.13 MHz for proton and equipped with a TCI cryoprobe. Deuterated water was used as the internal lock. The data have been processed with Bruker TOSPIN 3.0 software with standard parameter set. Phase and baseline correction were performed manually over the entire range of the spectra and the δ scale was calibrated to 0ppm using the internal standard TMSP.

## Methylglyoxal measurement in breast cancer cell culture media and pellets

MDA-MB-231 cells were cultured in low and high glucose conditions or were depleted for *GLO1* expression using siRNAs as described above. Forty-eight hours culture media were collected and the corresponding attached cells were scraped and counted to normalize methylglyoxal (MG)

measurements. Supernatants and cell pellets were snap-frozen and kept at −80°C until analysis. Levels of MG were determined in conditioned medium and cells by derivatization with O-phenylenediamine (oPD) and analyzed by stable isotope dilution ultra-performance liquid chromatography tandem mass spectrometry (UPLC-MS/MS) as described previously (*Scheijen and Schalkwijk, 2014*). Briefly, 30 µl of culture medium or cell lysate were mixed with 90 µl oPD (10 mg oPD in 10 ml 1.6 mol/l perchloric acid) in an Eppendorf cup. After an overnight (20 hr) reaction at room temperature and shielded from light, 10 µl of internal standard solution was added. Samples were mixed and subsequently centrifuged for 20 min at 21,000 g at a temperature of 4°C; 10 µl were injected for UPLC/MS/MS analysis.

## Glo1 activity assay

The activity of Glo1 was performed as previously described [*Chiavarina et al., 2014*]. Briefly, proteins were extracted with RIPA buffer, quantified and mixed with a pre-incubated (15 min at 25°C) equimolar (1 mM) mixture of MG and GSH (Sigma-Aldrich) in 50 mM sodium phosphate buffer, pH6.8. S-D-lactoylglutathione formation was followed spectrophotometrically by the increase of absorbance at 240 nm at 25°C. Glo1 activity data are expressed as arbitrary units (A.U.) of enzyme per mg of proteins. Three technical replicates of each sample have been performed and data are presented as mean ± SEM of five biological replicates.

## Chromatin immunoprecipitation (ChIP)

Formaldehyde was added directly to cell culture media to a final concentration of 1% at RT. Ten minutes later, glycine was added to a final concentration of 0.125M for 5 min at RT. The cells were then washed with ice-cold PBS, scraped, and collected in cold PBS followed by extraction in cell lysis buffer (20 mM Tris/HCl pH8, 85 mM KCl, 0.5% NP-40, protease inhibitor). Nuclei were pelleted by centrifugation at 2600g for 5 min at 4°C, suspended in nuclei lysis buffer (50 mM Tris/HCl pH8, 10 mM EDTA, 1% SDS, protease inhibitor) and sonicated with Bioruptor (Diagenode, Liège, Belgium). Samples were centrifuged at 14000 g for 15 min at 4°C. Supernatant were diluted in ChIP dilution buffer (0.01% SDS, 1.1% Triton X-100, 1.1 mM EDTA, 20 mM Tris/HCl pH8, 167 mM NaCl, protease inhibitor) to obtain a SDS final concentration of 0.2% and incubated with anti-YAP antibody (Santa Cruz, H125) or rabbit control IgG (Zymed Laboratories, ThermoFisher Scientific) overnight at 4°C. Protein G magnetic beads were blocked with BSA 0.1 mg/ml and salmon sperm DNA 0.1 mg/ml overnight at 4°C and then washed with ChIP dilution buffer. Beads were added to the lysate and incubated under rotation at 4°C. Four hours later, the beads were washed with low (0.1% SDS, 1% Triton X-100, 2 mM EDTA, 20 mM Tris/HCl pH8, 150 mM NaCl), high (0.1% SDS, 1% Triton X-100, 2 mM EDTA, 20 mM Tris/HCl pH8, 450 mM NaCl) salt wash buffer and LiCl wash buffer (0.5M LiCl, 1% NP-40, 1% deoxycholate, 20 mM Tris/HCl pH8). Next, the beads were incubated in elution buffer (50 mM NaHCO$_3$, 1% SDS) during 20 min under agitation. NaCl was added to a final concentration of 0.2M and samples were heated at 67°C overnight to reverse crosslinking. DNA was purified by phenol/chloroform extraction. The ChIP-enriched DNA was subjected to qPCR using connective tissue growth factor (CTGF) promoter TEAD binding-site-specific primers sense, 5′-ATATGAATCAG-GAGTGGTGCG-3′ and antisense, 5′-CAACTCACACCGGATTGATCC-3′ (*Fujii et al., 2012*). Primers (sens, 5′-AGACAAACCAAATCCAATCCACA-3′, antisens, 5′-CTGTGTTGGGTAGGTAGGGG-3′) targeting a more distal region on CTGF promoter were used as negative control. All qPCR data are normalized to Input and IgG controls and are presented as mean ± SEM of three biological replicates.

## Wound closure migration assay

MDA-MB-468 cells were transfected with 2 siRNAs against YAP and were grown to high density with MG 300 µM treatment. Multiple uniform streaks were made on the monolayer culture with 10 µl pipette tips. The cells were then washed to remove detached cells. Immediately after wounding and 16 hr later, each wound was photographed under a phase-contrast microscope. The distance between the wound edges was measured. Mean wound width was determined and a wound closure percentage was calculated for each condition. Sixteen wounds were measured per condition and the experiment was repeated twice. Data are expressed as the mean ± SEM.

## Cell growth assay

Equal numbers of cells were seeded, transfected with two siRNAs targeting YAP and treated with MG until confluence. Cell number was indirectly assessed using Hoechst incorporation at the indicated time period and cell growth was expressed based on cellular DNA content (µg/ml). Three technical replicates of each samples have been performed and data are presented as mean ± SEM of four biological replicates.

## Mass spectrometry – MG adducts localization

As previously described (*Dobler et al., 2006*; *Ahmed and Thornalley, 2005*; *Lund et al., 2011*), 5 µg of human recombinant Hsp90α (rhHsp90, Enzo Life Sciences, ADI-SPP-76D) were minimally modified with MG 500 µM in PBS 100 mM pH7.4 at 37°C during 24 hr. Proteins were reduced and alkylated, placed in 50 mM ammonium bicarbonate (buffer exchange was performed using an Amicon-3k from Millipore, Darmstadt, Germany) and then digested using a protease mixture. Peptides (15 pmoles injected) were separated by reverse phase chromatography (UPLC Waters nanoAcquity) in one dimension on a BEH C18 analytical column (25 cm length, 75 µM ID) with an increasing ratio of acetonitrile/water (5–40% in 85 min) at a 250 nl/min flow rate. The chromatography system was coupled to a hybrid Quadrupole-Orbitrap Mass Spectrometer (Q Exactive, ThermoFisher Scientific), operated in data-dependent acquisition mode. Survey scans were acquired at 70,000 resolving power (full width at half maximum, FWHM). Mass range was set from 400 to 1750 m/z in MS mode, and 1E6 ions were accumulated for the survey scans. After each survey scan, the 10 most intense ions were selected to do MS/MS experiments. Higher energy Collision Dissociation (HCD) fragmentation was performed at NCE 25 after isolation of ions within 2amu isolation windows. A dynamic exclusion of 10 s was enabled. Database searches were performed using Proteome Discoverer 1.4 (Thermo Scientific) in a Swissprot database (2014–05, 20339 human sequences) restricted to human taxonomy. MS and MS/MS tolerances were respectively set at 5 ppm and 20 mmu. Argpyrimidine (+80.026 Da, R), hydroimidazolone (+54.010 Da, R), dihydroxyimidazolidine (+72.021 Da, R) and carboxyethyllysine (+72.021 Da, K) were set as variable modifications while carbamidomethylation (+57.021 Da, C) was set as fixed modification.

## Mass spectrometry – MG adducts detection on endogenous Hsp90

Based on the experiments conducted using rhHsp90, a targeted method was set up to reach enough sensitivity to detect endogenous Hsp90 adducts in MDA-MB-231 MG-treated cells. Modified rhHsp90 as described above was first digested using Lys-C protease (in Tris-HCl 25 mM, pH8.5, 1 mM EDTA overnight at 37°C; first step at 1/40 sample/protease and then addition of 1/50 sample/protease in 50% acetonitrile for 4 hr). Resulting peptides were separated by reverse phase chromatography (UPLC Waters M Class) in one dimension on a HSS T3 C18 analytical column (25 cm length, 75 µM ID) with an increasing ratio of acetonitrile/water (2–40% in 32 min) at a 600 nL/min flow rate. The system was coupled to the mass spectrometer described above. A shortlist of 54 peptides were manually selected and used in further targeted experiments. Two 'Parallel Reaction Monitoring' or PRM (i.e. targeted full MS/MS) methods were set up in order to obtain at least 12 data points in chromatographic peaks and they were run consecutively for each sample. Data were then analyzed using Skyline 3.1 and were manually validated. For these experiments, protein extract from MDA-MB-231 treated or not with MG 300 µM during 6 hr were immunoprecipitated with argpyrimidine antibody. These samples were prepared in a slightly different way than rhHsp90: whole samples were reduced, alkylated and then purified using the 2D Clean-up kit (GE Healthcare, Milwaukee, WI, USA). The samples were then resuspended in the proteolysis buffer and the digestion was performed assuming an amount of 5 µg to be digested. The following steps were the same as for the recombinant protein.

## Immunoprecipitation and co-immunoprecipitation

MDA-MB-231 were treated with MG 300 µM during 6 hr. Then, argpyrimidine (mAb6B) and Hsp90 (anti-Hsp90 antibody, ab13492, Abcam, Cambridge, UK) and mouse IgG (Zymed Laboratories) immunoprecipitations were performed using the 'Crosslink IP' kit (#26147, ThermoFischer Sicentific) according to manufacturer instructions. For LATS1/Hsp90 co-immunoprecipitation, MDA-MB-231 were treated with MG 300 µM during 24 hr. Proteins were extracted in Tris-HCl pH8 20 mM, NaCl

137 mM, NP-40 1%, EDTA 2 mM and protease inhibitors. After incubation under rotation at 4°C during 30 min, cell lysates were centrifuged at 14,000g for 15 min at 4°C. Five hundred µg of proteins were incubated with 2 µg of LATS1 (Bethyl, Montgomery, TX, USA) or rabbit IgG (Zymed Laboratories) antibodies overnight and then 2 hr with Protein G magnetic beads at 4°C. After several washes, proteins were eluted and analyzed by Western blot. A representative western blot of three independent biological replicates is shown.

## Plasmids

pcDNA3.1-LATS-3xFlag and pcDNA3.1-3xFlag (empty vector) were kindly provided by Prof. Xiao-long Yang, Department of Pathology and Molecular Medicine, Queen's University, Kingston, Ontario K7L 3N6, Canada (*Hao et al., 2008*). Cell transfection was performed using Lipofectamine (Thermo-Fisher Scientific) according to manufacturer's instructions.

## Hsp90 ATPase activity

Hsp90 ATPase activity assay was performed as previously described (*Rowlands et al., 2010*) using Transcreener ADP[2] FI assay (BellBrook Labs, Fitchburg, WI, USA). Briefly, 1 µM of rhHsp90 was pre-incubated with MG 500 µM ± carnosine 10 mM or 17-AAG 1 µM during 24 hr at 37°C in Hepes pH7.4 50 mM, KCl 20 mM, EGTA 2 mM, $MgCl_2$ 4 mM and Brij-35 0.01%. ATP was added at a final concentration of 100 µM and incubated 3 hr at 37°C. The reaction was stopped and ADP was detected by adding ADP[2] Antibody-IRDye QC-1 at a final concentration of 93.7 µg/ml and ADP Alexa594 Tracer at a final concentration of 4 nM. This mix was incubated 1 hr at RT in a 96-well black half area plates (Greiner, Vilvoorde, Belgium, #675076). Readings were performed on a Filter Max F5 plate reader (Molecular Devices, Sunnyvale, CA, USA). Three technical replicates of each sample have been performed and data are presented as mean ± SEM of five biological replicates.

## In vivo mice experiments

All animal experimental procedures were performed according to the Federation of European Laboratory Animal Sciences Associations (FELASA) and were reviewed and approved by the Institutional Animal Care and Ethics Committee of the University of Liege (Belgium). Animals were housed in the GIGA-accredited animal facility of the University of Liege. For human xenografts, MDA-MB-231 shNT, sh*GLO1*#1 and sh*GLO1*#2 cells were suspended in 10% FBS supplemented medium and Matrigel (BD Biosciences) (50% v/v). Cell suspensions ($10^6$ cells/100 µl) were inoculated subcutaneously in one flank of 5-week-old female NOD-SCID mice (n = 15 per condition). After 4 weeks, tumors were surgically removed, weighted and measured with a caliper. Tumor volume (V) was assessed using the formula $V = \frac{4}{3} \times \pi \times \frac{H}{2} \times \frac{L}{2} \times \frac{W}{2}$ where H, L and W denote height, length and width, respectively. One piece was collected and embedded in paraffin for IHC and the rest was frozen in liquid nitrogen for total protein extraction. The animals were sutured, allowed to recover and further monitored for 6 weeks for metastases development. Due to animal ethics protocol, we had to sacrifice two mice in both sh*GLO1*#1 and sh*GLO1*#2 groups before the end of the experiment. A parallel experiment was conducted on sh*GLO1*#1 mice (n = 10) where they received carnosine (10 mM) in drinking water refreshed every 3 days from the day of primary tumor removal until the end of the experiment (for 6 weeks). Drinking volume was monitored and found to be similar between treated and non-treated mice. The mice were sacrificed and lung metastases were collected and processed as described for the primary tumors.

## Chicken chorioallantoic membrane (CAM) tumor assay

*GLO1*-depleted MDA-MB-231 cells were transfected with 2 different siRNAs directed against YAP (siYAP#1 and #2). On chicken embryonic day 11, 100 µl of a suspension of $2 \times 10^6$ (*Thornalley, 2005*) of MDA-MB-231 cells in culture medium mixed (1:1) with Matrigel (BD Biosciences) were deposited in the center of a plastic ring on the chicken embryo chorioallantoic membrane (n = 5). Tumors were harvested on embryonic day 18 and were fixed in 4% paraformaldehyde solution (30 min) for IHC analysis. Tumor volume was measured using a caliper and assessed using the formula described above. Parallel cultures of transfected cells were used to assess by Western blot that YAP silencing was maintained for the entire duration of the CAM assay experiment.

## Correlation analysis using YAP activity signature

The signature of YAP-modulated genes was described in previous studies (*Zhao et al., 2008*; *Cordenonsi et al., 2011*; *Dupont et al., 2011*; *Zhang et al., 2009*) and their mRNA levels were correlated to *GLO1* gene expression using publicly available GDS4057 dataset of 103 breast cancer patients (*Iwamoto et al., 2011*).

## Statistical analysis

Both technical and biological replicates were performed where indicated in figure legends. Technical replicates are considered as taking one sample and analyzing it several times in the same experiment. Biological replicates represent the analysis of samples from independent experiments. All results were reported as means with standard deviation (SD) or Standard Error Mean (SEM) as indicated in figure legends. Two group comparisons were performed using unpaired student's t-test with or without Welsch's correction according to homoscedasticity. When an experiment required comparisons between more than two groups, statistical analysis was performed using one-way or two-way ANOVA depending on the number of grouping factors. Dunnet's test was applied for simple comparisons while Student-Newman-Keul's (one-way ANOVA) or Bonferroni's (two-way ANOVA) tests were used for multiple comparisons. In the case of discrete variables (IHC scores) or non-normally distributed variables, the comparison between groups was performed by Mann-Whitney's U test, Wilcoxon Rank Sum test or a Kruskal-Wallis ANOVA followed by a Dunn's test according to the number of groups. Correlation between scores was assessed by the Spearman's rank correlation coefficient ($R_{spearman}$) and correlation between continuous variables was assessed by a Pearson correlation coefficient ($R_p$). Outliers were detected using whisker box plots. In all cases, a bilateral $p<0.05$ was considered as statistically significant with a 95% confidence interval. All experiments were performed as several independent biological replicates.

## Acknowledgements

MJN, FD and BCo are Télévie Fellows, BCh, ABl and AT are Télévie Post-Doctoral Fellows, PDT and ABe are Senior Research Associates, all from the National Fund for Scientific Research (FNRS, Belgium). This work was also supported by the Centre Anti-Cancéreux, University of Liège, Belgium. The authors are thankful to Mrs. N Maloujahmoum and Mr. V Hennequière for expert technical assistance. We acknowledge the technology platforms of the GIGA at University of Liège: proteomic, animal, imaging and flow cytometry, viral vector and immunohistology facilities. We thank the Tissue Bank of the University of Liège/University Hospital of Liège for providing human tumor samples.

## Additional information

### Funding

| Funder | Author |
| --- | --- |
| Université de Liège | Akeila Bellahcène |

MJN, FD and BCo are Télévie Fellows, BCh, ABl and AT are Télévie Post-Doctoral Fellows, PDT and ABe are Senior Research Associates, all from the National Fund for Scientific Research (FNRS, Belgium). This work was also supported by the Centre Anti-Cancéreux, University of Liège, Belgium. The funders had no role in study design, data collection and interpretation, or the decision to submit the work for publication.

### Author contributions

M-JN, Conception and design, Acquisition of data, Analysis and interpretation of data, Drafting or revising the article; FD, PP, BCh, OP, ABl, AT, BCo, NS, DBa, JLS, CGS, JL, PDT, EB, MT, EDP, PD, Acquisition of data, Analysis and interpretation of data, Drafting or revising the article; KU, DAS, JRC, CAH, DBe, Analysis and interpretation of data, Drafting or revising the article, Contributed unpublished essential data or reagents; VC, ABe, Conception and design, Analysis and interpretation of data, Drafting or revising the article

Author ORCIDs
James R Cochrane, http://orcid.org/0000-0002-8796-2143

## Ethics

Human subjects: Human breast tumor tissues were obtained from the Pathology Department of the University Hospital of Liege in agreement with ethical guidelines of the University of Liege, Belgium (#2015-155).

Animal experimentation: All animal experimental procedures were performed according to the Federation of European Laboratory Animal Sciences Associations (FELASA) and were reviewed and approved by the Institutional Animal Care and Ethics Committee of the University of Liege, Belgium (#14-1714). All surgery was performed under ketamin/xylazine anesthesia, and every effort was made to minimize suffering.

## Additional files

### Supplementary files

• Supplementary file 1. Antibodies and dilutions used for Western Blot experiments.

• Supplementary file 2. siRNA sequences.

• Supplementary file 3. Primer sequences and probes used for quantitative reverse transcription-PCR (qRT-PCR).

### Major datasets

The following previously published dataset was used:

| Author(s) | Year | Dataset title | Dataset URL | Database, license, and accessibility information |
|---|---|---|---|---|
| Iwamoto T, Bianchini G, Booser D, Qi Y, Coutant C, Shiang CY, Santarpia L, Matsuoka J, Hortobagyi GN, Symmans WF, Holmes FA, O'Shaughnessy J, Hellerstedt B, Pippen J, Andre F, Simon R, Pusztai L | 2011 | ER-positive/HER2-negative and ER-negative/HER2-negative breast cancer biopsies (MDACC/IGR cohort) | http://www.ncbi.nlm.nih.gov/sites/GDSbrowser?acc=GDS4057 | Publicly available at the NCBI Gene Expression Omnibus (accession no: GDS4057) |

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
