## [Decision Letter]

Thank you for submitting your article "Methylglyoxal, a glycolysis side-product, induces Hsp90 glycation and YAP-mediated tumor growth and metastasis" for consideration by *eLife*. Your article has been favorably evaluated by Tony Hunter (Senior Editor) and three reviewers, one of whom, Ralph DeBerardinis (Reviewer #1), is a member of our Board of Reviewing Editors, and another one is Jason W Locasale (Reviewer #3).

The reviewers have discussed the reviews with one another and the Reviewing Editor has drafted this decision to help you prepare a revised submission.

Summary:

The manuscript studies the role of methylglyoxal-mediated protein modification – a form of carbonyl stress associated with glycolysis – in oncogenic signaling. It demonstrates that methylglyoxal (MG) post-translationally modifies Hsp90, resulting in enhanced degradation of the tumor suppressor kinase LATS1. This enables persistent nuclear localization and transcriptional activity of YAP, leading to enhanced cell proliferation and migration. Increasing access to glucose was sufficient to increase MG modifications and YAP nuclear localization in breast cancer cells, and increasing MG levels by silencing glyoxylase-1 (Glo-1) had similar effects. In xenografts, chronic Glo1 silencing enhanced tumor growth and Ki67 content.

Essential revisions:

1) While it is clear from fluorescent images and western blots that MG induces an overall increase in YAP expression, it is not as clear from the ImageJ analysis that YAP is accumulating in the nucleus. As the paper stands, the data do not readily distinguish whether increased YAP is due to defective Hippo signaling per se, or a Hippo-independent mechanism(s) that leads to YAP stabilization. The authors should consider the following experiments to address this issue:

A) Examine TAZ in parallel with YAP, since a block in Hippo signaling should stabilize both YAP and TAZ.

B) Examine YAP S381 phosphorylation in parallel with S127 phosphorylation, with and without MG treatment, since a block in Hippo signaling should decrease both phosphorylation events.

2) MG levels are either measured indirectly with a fluorescent sensor (MBo) or directly using LC/MS. Absolute quantitation by LC/MS is helpful because it allows the reader to gauge the relevance of MG supplementation experiments, which are used throughout the paper. However, in high/low glucose (Figure 2) and Glo1 silencing (Figure 3) experiments, the authors only present data for extracellular MG levels, whereas the intracellular levels are more relevant to carbonyl stress. Intracellular MG levels should be quantified, and the authors should compare levels achieved by addition of exogenous MG addition to those achieved by high-glucose culture and Glo1 silencing. Otherwise this could represent a phenomenon related to supraphysiological MG levels with little physiological relevance.

3) The carnosine experiments to scavenge MG are helpful, but a better loss of function experiment would be to treat cells with an inhibitor of glycolysis (e.g. 2-deoxyglucose) that reduces the levels of glycolytic intermediates and MG. That would provide a better complement to the high/low glucose experiments than using extremely high levels of carnosine.

4) What happens to LATS2 under high-glucose conditions? Hsp90 is known to bind both LATS1 and LATS2, and these two kinases are redundant in Hippo signaling. Most likely LATS2 is also destabilized under hyperglycemia. The authors should examine this possibility along with a control kinase in the Hippo pathway such as MST1/MST2 under similar conditions.

---

## [Author Response]

*[…] Essential revisions:*

*1) While it is clear from fluorescent images and western blots that MG induces an overall increase in YAP expression, it is not as clear from the ImageJ analysis that YAP is accumulating in the nucleus. As the paper stands, the data do not readily distinguish whether increased YAP is due to defective Hippo signaling per se, or a Hippo-independent mechanism(s) that leads to YAP stabilization. The authors should consider the following experiments to address this issue:*

*A) Examine TAZ in parallel with YAP, since a block in Hippo signaling should stabilize both YAP and TAZ.*

We thank the reviewers for this constructive remark. To address this request, we have performed an immunoblot for TAZ in MDA-MB-231, MDA-MB-468 and MCF7 breast cancer cell lines treated with MG 300µM as shown now in Figure 2 and Figure 2—figure supplement 1. We also performed immunofluorescence experiments for TAZ in the three breast cancer cell lines analyzed treated with MG (Figure 2—figure supplement 3) and cultured in low and high glucose containing medium (Figure 3—figure supplement 2) and in stably Glo1-depleted MDA-MB-231 cells (Figure 4—figure supplement 1). Altogether the results demonstrated an accumulation of TAZ in breast cancer cells under all three carbonyl stress conditions under which we have previously demonstrated a YAP accumulation. This new information is now mentioned in the Results and Discussion.

*B) Examine YAP S381 phosphorylation in parallel with S127 phosphorylation, with and without MG treatment, since a block in Hippo signaling should decrease both phosphorylation events.*

We agree with this sound suggestion and have, accordingly, analyzed the phosphorylation of YAP on S381 in MDA-MB-231, MDA-MB-468 and MCF7 cells treated with MG using western blotting. In good accordance with a block in Hippo signaling, we observed a decrease of S381 phosphorylation as shown now in Figure 2 and Figure 2—figure supplement 1. This new information is now mentioned in the Results.

*2) MG levels are either measured indirectly with a fluorescent sensor (MBo) or directly using LC/MS. Absolute quantitation by LC/MS is helpful because it allows the reader to gauge the relevance of MG supplementation experiments, which are used throughout the paper. However, in high/low glucose (Figure 2) and Glo1 silencing (Figure 3) experiments, the authors only present data for extracellular MG levels, whereas the intracellular levels are more relevant to carbonyl stress. Intracellular MG levels should be quantified, and the authors should compare levels achieved by addition of exogenous MG addition to those achieved by high-glucose culture and Glo1 silencing. Otherwise this could represent a phenomenon related to supraphysiological MG levels with little physiological relevance.*

We thank the reviewers for this suggestion. As proposed, we have undertaken the LC/MS/MS measure of intracellular MG levels on MDA-MB-231 cell pellets cultured under high/low glucose, Glo1 silencing as well as upon exogenous MG addition. For this purpose, we have asked for the contribution of external collaborators (J.S and C.S, now added as co-authors) who had the expertise for MG dosage on cell pellets using UPLC/MS/MS. To homogenize the techniques used, extracellular MG levels were measured in parallel on matched supernatant samples. Therefore, new extracellular MG concentrations expressed as nanomoles/L normalized to the number of cells are shown in Figure 3 and Figure 3—figure supplement 1. Intracellular MG concentrations measured on cell pellet lysates are shown below in Figure 10. Several conclusions can be drawn from these measurements: (1) Intracellular MG concentrations are more elevated than extracellular ones. This observation is in good accordance with previous studies (Rabbani and Thornalley, Nature Protocols, 2014) and has been notably attributed to the fact that intracellular concentrations account for both free MG and reversibly bound MG to cellular targets (proteins, nucleic acids and lipids) (Chaplen et al., Analytical Biochemistry, 1996). (2) As shown in Figure 10, intracellular MG concentrations levels were found to be very similar in all conditions indicating that MG quantification in cells is not relevant to assess the extent of carbonyl stress. In fact, such measurement of MG has never been reported before in cancer cells and several questions await further detailed analytical studies. Is free MG gradient the factor that equilibrates its concentration between extracellular and intracellular compartments? The data shown tend to indicate that it is not the case as 6-fold more MG is found in the extracellular medium than in cells upon exogenous MG treatment (300μm 48h). According to its high reactivity, is it conceivable that free MG could be measured inside the cells? No doubt that Glo1 activity level will also control final free MG concentrations. (3)It is particularly striking that under exogenous MG stimulation, MG concentration inside the cells remains stable while it is significantly increased in the extracellular medium (Figure 10). An effect that could be attributed to the presence of cellular Glo1 activity that maintains MG levels under a certain sub-cytotoxic threshold. In good accordance with this hypothesis, we have previously demonstrated that MDA-MB-231 cells increase their Glo1 activity in response to exogenous MG addition (Chiavarina et al., Oncotarget, 2014). Based on these new experiments, we conclude that extracellular concentrations of MG are the most relevant to assess carbonyl stress in all 3 experimental conditions. When it comes to intracellular MG evaluation, we strongly believe that the quantification of MG-adducts using western blotting is a better reflection of carbonyl stress under all 3 experimental conditions (Figure 10). MG level in MDA-MB-231 cells was approximately 6μM which is slightly higher than the previous range of concentrations reported for normal mammalian cells (1 to 4 μM) which may be attributable to the glycolytic phenotype of these cancer cells. The role of MG-mediated carbonyl stress has been understudied in cancer but it is gaining much of interest nowadays (Sullivan LB, et al., Nature reviews Cancer, Sept. 2016). To the best of our knowledge MG concentrations have not yet been established in cancer patients in comparison with healthy subjects. Moreover, the local concentrations of MG achieved in highly glycolytic tumors are not known and must be a balance between the amount produced (glycolytic flux), amount carried away by vasculature and how it is catabolized by tumor and/or stromal cells. Our in vitro studies indicate that MDA-MB-231 cells can tolerate high amounts of MG (IC50 = 938μM). Until more studies are undertaken in this direction it is difficult to assess what should be considered as supraphysiological levels of MG and pending a better understanding of how and what proportion of exogenous MG enters the cells, the detection of MG adducts accumulation and/or extracellular MG concentrations in tumor cells represents the best indication of MG-mediated carbonyl stress upon both exogenous and endogenous MG supply in our in vivo and in vitro experiments.

Author response image 1.Intracellular (**A**) and extracellular (**B**) MG concentrations in MDA-MB-231 cells.MG adducts western blotting detection using anti-argpyrimidine antibody in MDA-MB-231 cells cultured in high and low glucose and upon exogenous MG treatment (**C**) and in Glo1-silenced cells (**D**).**DOI:**
http://dx.doi.org/10.7554/eLife.19375.030

*3) The carnosine experiments to scavenge MG are helpful, but a better loss of function experiment would be to treat cells with an inhibitor of glycolysis (e.g. 2-deoxyglucose) that reduces the levels of glycolytic intermediates and MG. That would provide a better complement to the high/low glucose experiments than using extremely high levels of carnosine.*

We thank the reviewers for this suggestion. Therefore, we performed new experiments where MDA-MB-231 and MDA-MB-468 cells cultured in high-glucose medium were treated with 2-deoxyglucose (2-DG) 5mM during 48h. We observed a significant decrease of lactate and MG production compared to control in glycolytic MDA-MB-231 and MDA-MB-468 cells as shown in the new Figure 3—figure supplement 3. As expected, we found a decreased YAP nuclear immunodetection in glycolytic breast cancer cells treated with 2-DG (Figure 3—figure supplement 3).

*4) What happens to LATS2 under high-glucose conditions? Hsp90 is known to bind both LATS1 and LATS2, and these two kinases are redundant in Hippo signaling. Most likely LATS2 is also destabilized under hyperglycemia. The authors should examine this possibility along with a control kinase in the Hippo pathway such as MST1/MST2 under similar conditions.*

We thank the reviewers for this pertinent comment which prompted us to evaluate LATS2, Mst1 and Mst2 expression levels in MDA-MB-231, MDA-MB-468 and MCF7 cells under the same condition that induced a decrease of LATS1 expression. As shown in Figure 6, we observed no change in LATS2 expression in the 3 breast cancer cell lines indicating that LATS2 expression did not obey to the same regulation as LATS1 upon MG treatment.

When we checked for MST1 and MST2 expression in the 3 cell lines under the same treatment, we observed a stable (MDA-MB-468 and MCF7) and even an increased (MDA-MB-231) expression of these kinases indicating that MG stress did not affect negatively the Hippo pathway upstream of LATS1 kinase level. These results are shown in Figure 6 in the new version of the manuscript and mentioned in the Discussion.